# Epigenetic regulation of *Wnt7b* expression by the *cis*-acting long noncoding RNA Lnc-Rewind in muscle stem cells

Andrea Cipriano[1†‡], Martina Macino[1,2†], Giulia Buonaiuto[1], Tiziana Santini[1,3], Beatrice Biferali[1,2], Giovanna Peruzzi[3], Alessio Colantoni[1], Chiara Mozzetta[2]*, Monica Ballarino[1]*

[1]Department of Biology and Biotechnology Charles Darwin, Sapienza University of Rome, Rome, Italy; [2]Institute of Molecular Biology and Pathology (IBPM), National Research Council (CNR) at Sapienza University of Rome, Rome, Italy; [3]Center for Life Nano Science at Sapienza, Istituto Italiano di Tecnologia, Rome, Italy

*For correspondence:
chiara.mozzetta@uniroma1.it
(CM);
monica.ballarino@uniroma1.it
(MB)

†These authors contributed
equally to this work

Present address: ‡Department
of Obstetrics & Gynecology,
Stanford University, Stanford,
United States

Competing interests: The
authors declare that no
competing interests exist.

Reviewing editor: Jeannie T
Lee, Massachusetts General
Hospital, United States

**Abstract** Skeletal muscle possesses an outstanding capacity to regenerate upon injury due to the adult muscle stem cell (MuSC) activity. This ability requires the proper balance between MuSC expansion and differentiation, which is critical for muscle homeostasis and contributes, if deregulated, to muscle diseases. Here, we functionally characterize a novel chromatin-associated long noncoding RNA (lncRNA), Lnc-Rewind, which is expressed in murine MuSCs and conserved in human. We find that, in mouse, Lnc-Rewind acts as an epigenetic regulator of MuSC proliferation and expansion by influencing the expression of skeletal muscle genes and several components of the WNT (Wingless-INT) signalling pathway. Among them, we identified the nearby Wnt7b gene as a direct Lnc-Rewind target. We show that Lnc-Rewind interacts with the G9a histone lysine methyltransferase and mediates the in cis repression of Wnt7b by H3K9me2 deposition. Overall, these findings provide novel insights into the epigenetic regulation of adult muscle stem cells fate by lncRNAs.

## Introduction

The transcriptional output of all organisms was recently found to be more complex than originally imagined, as the majority of the genomic content is pervasively transcribed into a diverse range of regulatory short and long non-protein coding RNAs (ncRNAs) (*Abugessaisa et al., 2017*; *Carninci et al., 2005*; *Cipriano and Ballarino, 2018*). Among them, long noncoding RNAs (lncRNAs) are operationally defined as transcripts longer than 200 nucleotides, which display little or no protein coding potential. Since the initial discovery, numerous studies have demonstrated their contribution to many biological processes, including pluripotency, cell differentiation, and organism development (*Ballarino et al., 2016*; *Fatica and Bozzoni, 2014*). LncRNAs were demonstrated to regulate gene expression at transcriptional, post-transcriptional, or translational level. The function and the mechanisms of action are different and primarily depend on their (nuclear or cytoplasmic) subcellular localization (*Carlevaro et al., 2019*; *Rinn and Chang, 2012*; *Ulitsky and Bartel, 2013*). A large number of lncRNAs localize inside the nucleus, either enriched on the chromatin or restricted to specific nucleoplasmic foci (*Engreitz et al., 2016*; *Sun et al., 2018*). In this location, they have the capacity to control the expression of neighbouring (in cis) or distant (in trans) genes by regulating their chromatin environment and also acting as structural scaffolds of nuclear domains (*Kopp and Mendell, 2018*). Among the most important functions proposed for *cis*-acting lncRNAs, there is their ability to

regulate transcription by recruiting repressing or activating epigenetic complexes to specific genomic loci (*Huarte et al., 2010*; *Maamar et al., 2013*; *McHugh et al., 2015*; *Wang et al., 2011*).

In muscle, high-throughput transcriptome sequencing (RNA-seq) and differential expression analyses have facilitated the discovery of several lncRNAs that are modulated during the different stages of skeletal myogenesis and dysregulated in muscle disorders (*Ballarino et al., 2016*; *Lu et al., 2013*; *Zhao et al., 2019*). Although the roles of these transcripts have been partially identified, we are still far from a complete understanding of their mechanisms of action. For instance, the knowledge of the lncRNAs impact on adult muscle stem cell (MuSC) biology is partial and only a few examples have been characterized as functionally important. In the cytoplasm, the lncRNA *Lnc-mg* has been shown to regulate MuSC differentiation by acting as a sponge for *Mir125b* (*Zhu et al., 2017*). In the nucleus, the lncRNA *Dubr* was found to promote satellite cell differentiation by recruiting Dnmts to the developmental pluripotency-associated 2 (*Dppa2*) promoter, leading to CpG hypermethylation and silencing (*Wang et al., 2015*). In mouse, we have previously identified several lncRNAs specifically expressed during muscle in vitro differentiation and with either nuclear or cytoplasmic localization (*Ballarino et al., 2015*). Among them, we found Lnc-Rewind (**Re**pressor of **w**nt **ind**uction), a chromatin-associated lncRNA conserved in human and expressed in proliferating $C_2C_{12}$ myoblasts.

Here, we provide evidence on the role of Lnc-Rewind in the epigenetic regulation of the WNT (Wingless-INT) signalling in muscle cells. The WNT transduction cascade has been demonstrated to act as a conserved regulator of stem cell function via canonical (β-CATENIN) and non-canonical (planar cell polarity and calcium) signalling, and dysregulation of its activity has been reported in various developmental disorders and diseases (*Nusse and Clevers, 2017*). In the muscle stem cell niche, WNT signalling is key in coordinating MuSC transitions from quiescence, proliferation, commitment, and differentiation (*Brack et al., 2008*; *Eliazer et al., 2019*; *Lacour et al., 2017*; *Le Grand et al., 2009*; *Parisi et al., 2015*; *Rudolf et al., 2016*). Because of this central role, the WNT pathway is supervised by several mechanisms and different works have shown that lncRNAs can modulate it at both transcriptional and post-transcriptional levels (*Zarkou et al., 2018*). Here, we provide evidence that Lnc-Rewind associates with the H3K9 methyltransferase G9a to regulate the deposition of H3K9me2 in cis on the nearby *Wnt7b* gene. Our data show that Lnc-Rewind expression is necessary to maintain *Wnt7b* repressed and to allow MuSC expansion and proper differentiation.

## Results

### Lnc-Rewind is a conserved chromatin-associated lncRNA expressed in satellite cells

In an attempt to uncover novel regulators of MuSC activity, we decided to take advantage of the atlas of newly discovered lncRNAs, which we previously identified as expressed in proliferating muscle cells (*Ballarino et al., 2015*). Among them, we focused on Lnc-Rewind, which is a lncRNA enriched in proliferating myoblasts and overlapping *pre-Mirlet7c-2* (*mmu-let7c-2*) and *pre-Mirlet7b* (*mmu-let7b*) genomic loci (*Figure 1A*). An evolutionary conservation analysis performed by examining FANTOM5 datasets (*Noguchi et al., 2017*) revealed the existence of a conserved transcriptional start site (TSS) localized in the human (hs_Lnc-Rewind) syntenic locus (*Figure 1B*). This region exhibits an overall ~46% of (exonic and intronic) sequence identity (*Figure 1—figure supplement 1A*), which is relatively high for lncRNAs. Moreover, RNA-seq (*Legnini et al., 2017*; *Figure 1B*) and semi-quantitative (sq)RT-PCR analyses (*Figure 1—figure supplement 1B*) confirmed that, in proliferating human myoblasts, this region is actively transcribed. To note, the distribution of the read coverage in both mouse and human revealed the existence of spliced and unspliced Lnc-Rewind isoforms originating from the two loci (*Figure 1A,B*).

To pinpoint possible roles of the murine transcript in muscle cells, we first assessed Lnc-Rewind expression and subcellular localization in both $C_2C_{12}$ and Fluorescence-activated cell sorting (FACS) isolated MuSC (*Figure 1—figure supplement 1C*) grown under proliferative (hereafter referred to as MuSC-derived myoblasts) and differentiating conditions. Proper myogenic differentiation was confirmed by the expression of late muscle-specific genes such as myogenin (*Myog*) and muscle creatin kinase (*Mck*) (*Figure 1—figure supplement 1D*). Quantitative qRT-PCR analysis revealed that Lnc-Rewind expression is high in proliferating (GM) $C_2C_{12}$ and MuSC-derived myoblasts and significantly decreased in fully differentiated (DM) cells (*Figure 1C*). Subcellular

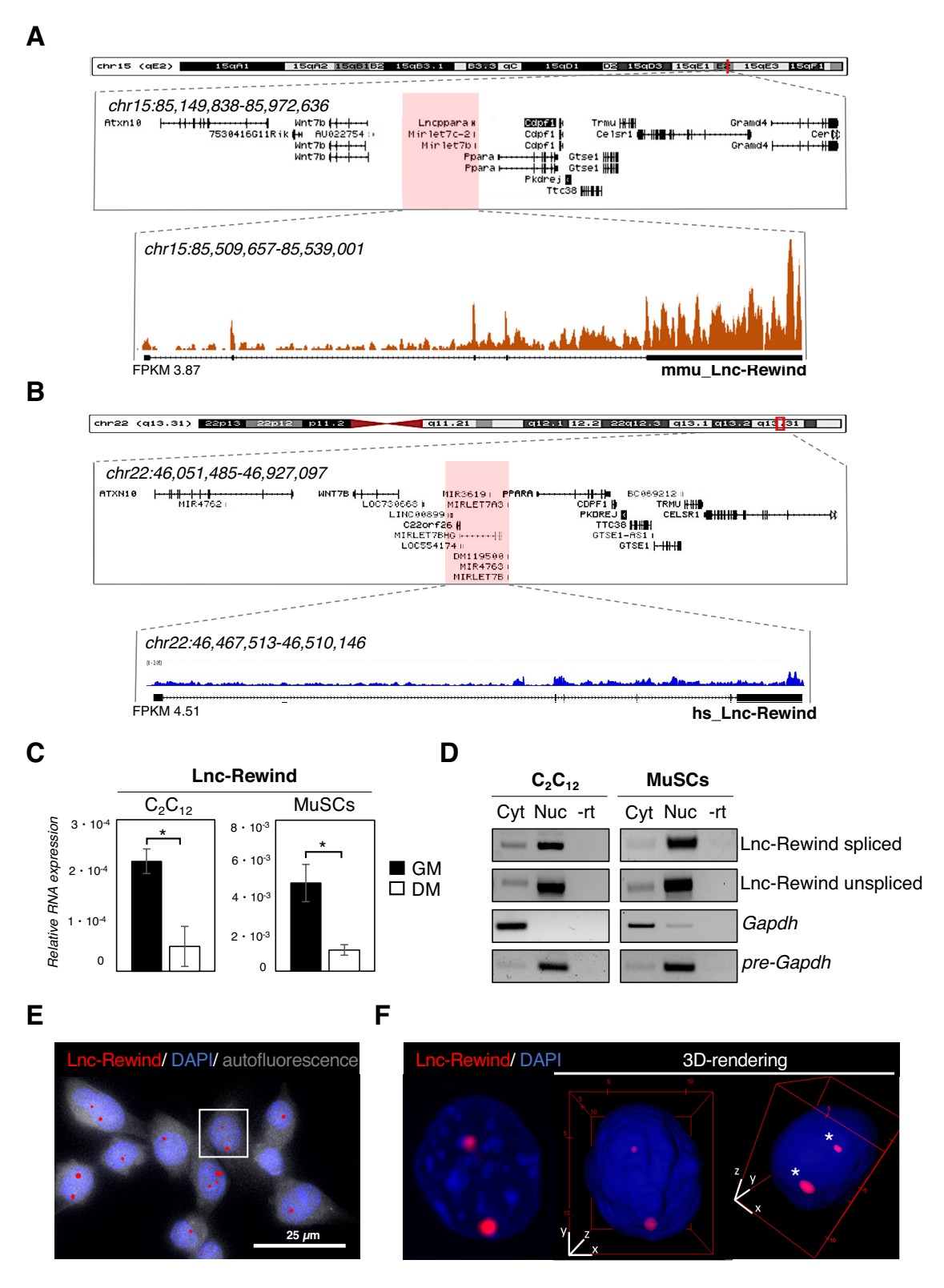

**Figure 1.** Lnc-Rewind is a conserved chromatin-associated lncRNA expressed in satellite cells. (**A**) UCSC visualization showing the chromosome position and the genomic coordinates of Lnc-Rewind (red shade) in the mm9 mouse genome. Mmu_Lnc-Rewind reads coverage and quantification (FPKM) from RNA-Seq experiments performed in proliferating $C_2C_{12}$ cells (*Ballarino et al., 2015*; GSE94498) are shown. (**B**) UCSC visualization showing the chromosome position and the genomic coordinates of hs_Lnc-Rewind (red shade) in the hg19 human genome. Hs_Lnc-Rewind reads coverage and

*Figure 1 continued on next page*

*Figure 1 continued*

quantification (FPKM) from RNA-Seq experiments performed in proliferating myoblasts (*Legnini et al., 2017*; GSE70389) are shown together with the genomic structure of the human locus (magnified box). (**C**) Relative Lnc-Rewind expression assessed by quantitative RT-PCR (qRT-PCR) in $C_2C_{12}$ myoblast ($C_2C_{12}$) and MuSC-derived myoblasts (referred as MuSCs) maintained in growing (GM) or differentiated (DM) conditions. Data represent the mean ± SEM from three biological replicates. *Gapdh* RNA was used as a normalization control. (**D**) Semiquantitative RT-PCR (sqRT-PCR) analysis of spliced and unspliced Lnc-Rewind isoforms in cytoplasmic (Cyt) and nuclear (Nuc) fractions from proliferating $C_2C_{12}$ and MuSC-derived myoblasts. The quality of fractionation was tested with mature (*Gapdh*) and precursor (*pre-Gapdh*) RNAs. –rt represents the negative control. (**E**) RNA-FISH analysis for Lnc-Rewind RNA (red) in proliferating MuSC-derived myoblasts. Autofluorescence (grey) is shown with false colour to visualize the cell body. (**F**) Digital magnification and 3D visualization of the square insert of (**E**). Asterisks indicate the Lnc-Rewind RNA signals inside the nuclear volume. DAPI, 4',6-diamidino-2-phenylindole (blue). Data information: *p<0.05, unpaired Student's t-test.

The online version of this article includes the following source data and figure supplement(s) for figure 1:

**Source data 1.** Source data for *Figure 1*.
**Figure supplement 1.** Lnc-Rewind is a conserved chromatin-associated lncRNA expressed in satellite cells.
**Figure supplement 1—source data 1.** Source data for *Figure 1—figure supplement 1*.

fractionation of cytoplasmic (Cyt) and nuclear (Nuc) fractions showed that both spliced and unspliced isoforms localizes in the nuclear compartment (*Figure 1D*, *Figure 1—figure supplement 1E*). Accordingly, RNA fluorescence in situ hybridization (FISH) experiments performed in both MuSC-derived myoblasts (*Figure 1E,F*, *Figure 1—figure supplement 1F*) and $C_2C_{12}$ cells (*Figure 1—figure supplement 1G*) confirmed the nuclear localization of Lnc-Rewind and further revealed its specific enrichment to discrete chromatin foci. Overall, these results point towards a role for Lnc-Rewind in chromatin-based processes and suggest its possible involvement in the epigenetic regulation of MuSC homeostasis.

## Lnc-Rewind regulates muscle system processes and MuSC expansion

To gain insights into the functional role of Lnc-Rewind and to identify the molecular pathways involved in the regulation of muscle stem cells, we performed a global transcriptional profiling on MuSC-derived myoblasts treated with either a mix of three different LNA GapmeRs against Lnc-Rewind (GAP-REW) or the scramble control (GAP-SCR) (*Figure 2—figure supplement 1A*, upper panel). Under these conditions, we obtained ~70% reduction of Lnc-Rewind expression (*Figure 2—figure supplement 1A*, lower panel), which led to the identification of a set of 1088 differentially expressed genes (DEGs) (p<0.05, GAP-SCR vs GAP-REW). Of these, 332 were upregulated and 756 downregulated in GAP-REW as compared to the GAP-SCR condition (*Figure 2A*, *Figure 2—source data 1*). A principal component analysis (PCA) performed on the DEG datasets revealed that the GAP-SCR and GAP-REW experimental groups displayed a clear different pattern of gene expression since they occupy different regions of the PCA plot (*Figure 2—figure supplement 1B*). The DEG list was then subjected to Gene Ontology (GO) term enrichment analysis (Biological process) to define functional clusters. It emerged that DEGs were mostly associated with muscle cell physiology (skeletal muscle contraction, p-value=4.23E-6) (*Figure 2B*, *Figure 2—figure supplement 1C*). Of note, the analysis of Lnc-Rewind generated miRNAs (*Mirlet7b* and *Mirlet7c-2*) revealed that, although the lncRNA depletion results on their concomitant downregulation (*Figure 2—figure supplement 1D*), none of the upregulated transcripts that are also putative *Mirlet7b* and *Mirlet7c-2* targets (~1% of the upregulated genes) (*Figure 2—figure supplement 1E*), belong to any of the GO enriched categories. This result emphasizes a specific and miRNA-independent role for Lnc-Rewind. Both 'muscle system process' (p-value=3.45E-7) and 'striated muscle contraction' (p-value=4.63E-7) were the most significantly enriched GO terms (*Figure 2—figure supplement 1C*). Of note, genes encoding for different proteins involved in muscle contraction, such as myosins (i.e. *Myh8*, *Myl1*, *Myh3*) and troponins (i.e. *Tnnt2*, *Tnnt1*) (*Figure 2C*), were downregulated. Accordingly, Lnc-Rewind-depleted cells express lower levels of MyHC protein (*Figure 2D*). Moreover, morphological evaluation highlighted a decreased number of MuSC-derived myoblasts after Lnc-Rewind depletion (*Figure 2—figure supplement 1F*), suggesting a primary defect in MuSC proliferation. This led us to hypothesize that the defects in myogenic capacity (*Figure 2C,D*, *Figure 2—figure supplement 1F*) might result by a decreased cell density that normally leads to a lower rate of myogenic differentiation. In line with this hypothesis, EdU incorporation experiments revealed a striking reduction of proliferating MuSC-derived myoblasts upon depletion of Lnc-Rewind (*Figure 2E*). Accordingly, the *Ccnd3* gene

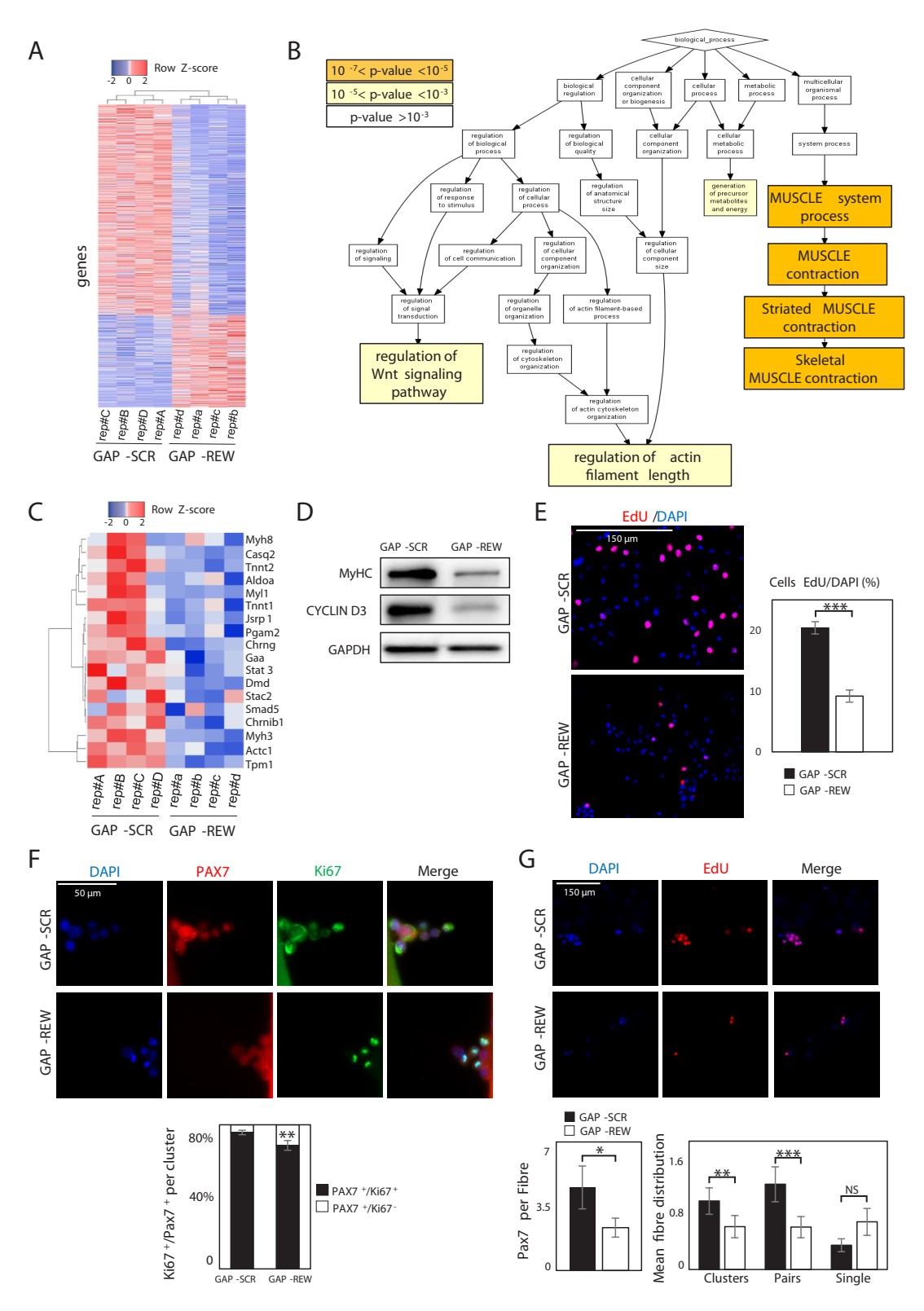

**Figure 2.** Lnc-Rewind regulates muscle system processes and MuSC expansion. (**A**) Heatmap representing hierarchical clustering performed on the final list of genes differentially expressed in MuSC-derived myoblasts upon Lnc-Rewind depletion (p-value threshold<0.05; see also *Figure 2—source data 1*). The analysis was performed using the Heatmapper webserver tool (*Babicki et al., 2016*). For each gene, expression levels are expressed as Z-score values. (**B**) Gene Ontology (GO) enrichment analysis performed by GORILLA (*Eden et al., 2009*) in Biological process for genes differentially expressed

*Figure 2 continued on next page*

*Figure 2 continued*

upon Lnc-Rewind depletion in MuSC-derived myoblasts. (C) Heatmap representing hierarchical clustering performed on DEGs belonging to the Skeletal Muscle contraction GO category (GO:0003009). The analysis was performed using the Heatmapper webserver tool (*Babicki et al., 2016*). For each gene, expression levels are expressed as Z-score values. (D) Western blot analysis performed on protein extracts from MuSC-derived myoblasts treated with GAP-SCR or GAP-REW. GAPDH protein was used as endogenous control. (E) Representative images of MuSC-derived myoblasts treated with GAP-SCR and GAP-REW and incubated with EdU (red). Nuclei were visualized with DAPI (blue). Histogram shows the percentage of EdU-positive cells on the total of DAPI-positive cells. Data are graphed as mean ± SEM; n = 6 mice. (F) Representative images of single muscle fibres from WT mice treated with GAP-SCR or GAP-REW and stained for Pax7 (red) and Ki67 (green). Nuclei were visualized with DAPI (blue); histogram shows the percentage of $Pax7^+/Ki67^-$ cells on the total of $Pax7^+/Ki67^+$ cells per cluster. Data represent the mean ± SEM of 40 clusters per condition; n = 5 mice. (G) Representative images of single muscle fibres from WT mice treated with GAP-SCR or GAP-REW and incubated for 24 hr with EdU (red). Nuclei were visualized with DAPI (blue); left histogram shows the number of $Pax7^+$ cells per fibre (50 fibres per condition; n = 5 mice). Right histograms represent the mean of clusters (nuclei [n], n > 2), pairs (n = 2), and single cells $Pax7^+$ (n = 1) per fibre. Data represent the mean ± SEM of 80 fibres per condition; n = 5 mice. Data information: *p<0.05, **p<0.01, ***p<0.001, unpaired Student's t-test.

The online version of this article includes the following source data and figure supplement(s) for figure 2:

**Source data 1.** List of the differentially expressed genes (DEGs) between GAP-SCR and GAP-REW transfected MuSCs.
**Source data 2.** Source data for *Figure 2*.
**Figure supplement 1.** Lnc-Rewind regulates muscle system processes and MuSC expansion.
**Figure supplement 1—source data 1.** Source data for *Figure 2—figure supplement 1*.

encoding for Cyclin D3, a cyclin specifically involved in promoting transition from G1 to S phase, was significantly downregulated in Lnc-Rewind-depleted cells at both transcript (*Figure 2—source data 1*) and protein levels (*Figure 2D*). In further support of this, MuSCs on single myofibres cultured for 96 hr gave rise to a decreased percentage of proliferating ($Pax7^+/Ki67^+$) progeny upon Lnc-Rewind downregulation (*Figure 2F*). Moreover, quantification of the number of $Pax7^+$-derived clusters (composed of more than two nuclei), pairs (composed of two nuclei), or single MuSCs within each myofibre revealed a reduction of activated pairs and clusters upon Lnc-Rewind depletion (*Figure 2G*), suggesting a role for the lncRNA in sustaining MuSC activation and expansion.

## Lnc-Rewind and *Wnt7b* genes display opposite pattern of expression and a functional interplay

Together with the muscle-specific genes, the 'regulation of Wnt signalling pathway' GO term caught our attention as it was represented by a significant subset of trancripts (p-value=5.44E-4, *Figure 2—figure supplement 1C*). Among them, we found *Wnt7b*, which expression was found upregulated at both transcript and protein levels (*Figure 3—figure supplement 1A* and *Figure 2—source data 1*), suggesting a role for Lnc-Rewind as a repressor of *Wnt7b* expression. Intriguingly, *Wnt7b* transcriptional locus localizes only 100 kb upstream Lnc-Rewind gene (*Figure 3A*).

The genomic proximity between Lnc-Rewind/*Wnt7b* loci, together with their anti-correlated expression in muscle cells and the lncRNA chromatin enrichment (*Figure 1D,E*), led us to hypothesize that Lnc-Rewind might have a direct, in cis-regulatory role on *Wnt7b* transcription. A first clue in favour of such hypothesis came from FANTOM5 Cap Analysis Gene Expression (CAGE) profiles of mouse samples, available on ZENBU genome browser (*Noguchi et al., 2017*). Indeed, the inspection of the TSS usage among all the available murine samples (n = 1196, *Figure 3—source data 1*) revealed a distinctive anti-correlated expression between Lnc-Rewind and *Wnt7b* transcripts (*Figure 3B*, left panels). In contrast, the other Lnc-Rewind-neighbouring genes, such as *Cdpf1* and *Atxn10*, displayed no specific expression correlation (*Figure 3B*, right panels). Of note, *Wnt7b* was the only gene that was significantly upregulated upon the lncRNA depletion with a concomitant increase in WNT7b protein levels (*Figure 3C*, *Figure 3—figure supplement 1A, B*).

To uniquely link the defects in MuSC activation/proliferation induced by Lnc-Rewind depletion (*Figure 2E–G*) to the aberrant induction of *Wnt7b* expression, we performed a rescue experiment. To this end, in a context where the upregulation of *Wnt7b* was triggered by Lnc-Rewind depletion, we restored its expression to physiological levels by si-*Wnt7b*-mediated knockdown (*Figure 3—figure supplement 1C*). Of note, rescuing *Wnt7b* repression promoted an increase in the percentage of proliferating MuSC-derived myoblasts towards control levels (*Figure 3D*). Similarly, restoration of *Wnt7b* repression on single myofibres was sufficient to restore the capacity of the associated MuSCs to properly activate and proliferate ex vivo (*Figure 3E*), as assessed by the quantification of the

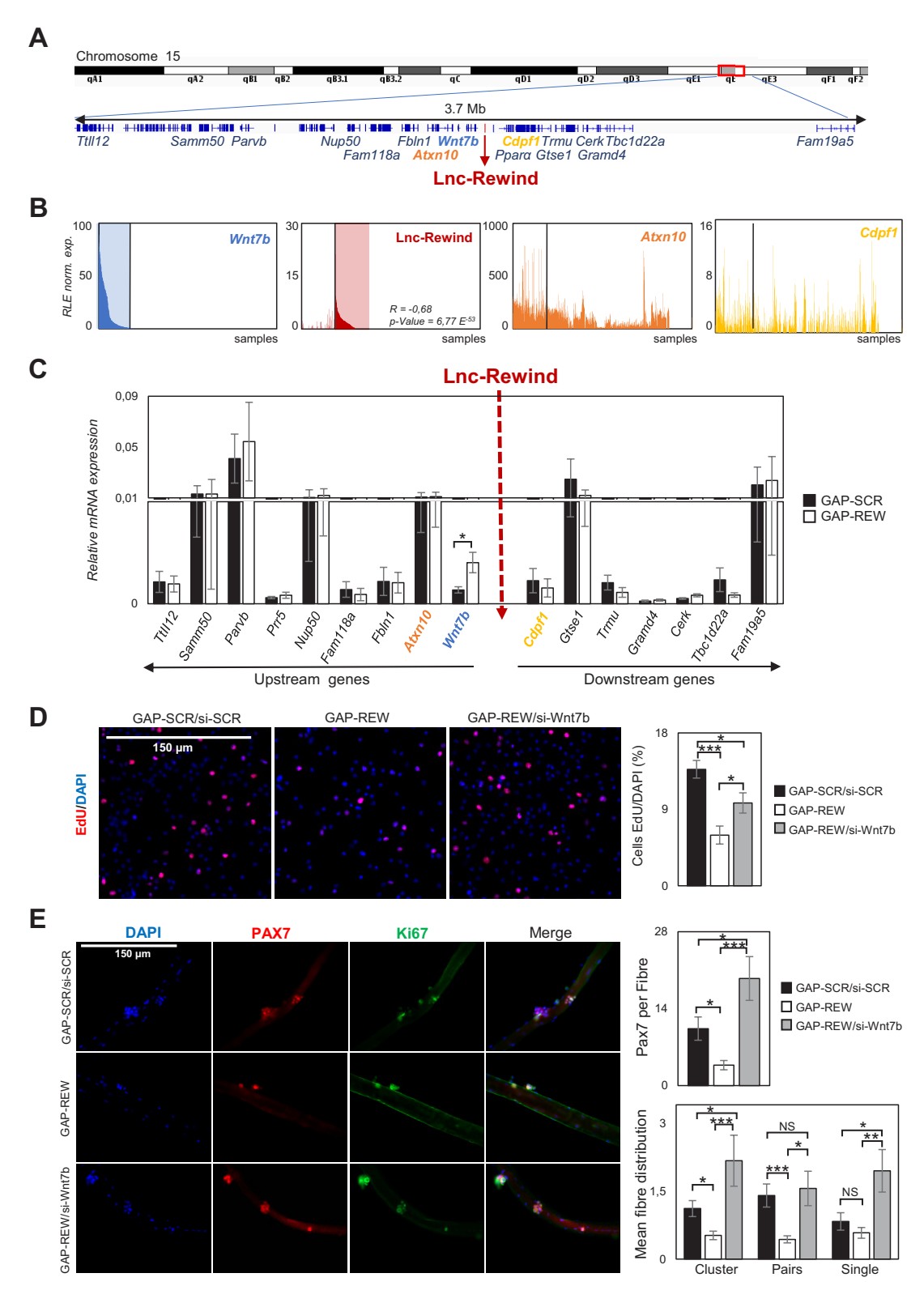

**Figure 3.** Lnc-Rewind and *Wnt7b* genes display opposite pattern of expression and a functional interplay. (**A**) Schematic UCSC visualization showing the Lnc-Rewind chromosome position and its neighbouring genes. (**B**) TSS usage analyses of *Wnt7b*, Lnc-Rewind, *Atxn10*, and *Cdpf1* performed by using FANTOM5 (Phases 1 and 2) CAGE datasets. Each bar represents the Relative Logarithmic Expression (RLE) of the tag per million (TPM) values of the TSS of each gene in one sample (1196 samples). The order of the samples is the same in each of the histograms and shown in *Figure 3—source*

*Figure 3 continued on next page*

*Figure 3 continued*

*data 1*. R value represents the Spearman's rank correlation coefficient between Lnc-Rewind and *Wnt7b* expression, and the p-value was calculated using the Spearman's rank correlation test. (**C**) qRT-PCR quantification of Lnc-Rewind neighbouring genes in GAP-SCR *versus* GAP-REW treated. MuSCs-derived myoblasts. Data were normalized to *Gapdh* mRNA and represent the average ± SEM from four biological replicates. (**D**) Representative images of MuSC-derived myoblasts treated with GAP-SCR/si-SCR, GAP-REW, and GAP-REW/si-Wnt7b and incubated with EdU (red). Nuclei were visualized with DAPI, 4′,6-diamidino-2-phenylindole (blue). Histogram shows the percentage of EdU-positive cells on the total of DAPI-positive cells. Data are graphed as mean ± SEM; n = 5 mice. (**E**) Representative images of single muscle fibres treated with GAP-SCR/si-SCR, GAP-REW, and GAP-REW/si-Wnt7b from WT mice after 96 hr in culture, stained for Pax7 (red) and Ki67 (green). Nuclei were visualized with DAPI, 4′,6-diamidino-2-phenylindole (blue). Histogram (upper panel) shows the number of Pax7$^+$ cells per fibre. Histograms (lower panel) represent the mean of the number of clusters (nuclei [n]: n > 2), pairs (n = 2), and single cells Pax7$^+$ (n = 1) per fibre. Data represent the mean ± SEM; n = 5 mice. Data information: (C): *p<0.05, paired Student's t-test. 3D and 3E: *p<0.05, **p<0.01, ***<0.001, one-way Anova with Tukey's multiple comparison test. The online version of this article includes the following source data and figure supplement(s) for figure 3:

**Source data 1.** 5′CAGE TSS expression profile of Wnt7b, Lnc-Rewind, Atxn10 and Cdpf1 genes.
**Source data 2.** Source data for *Figure 3*.
**Figure supplement 1.** Lnc-Rewind and *Wnt7b* genes display opposite pattern of expression and a functional interplay.
**Figure supplement 1—source data 1.** Source data for *Figure 3—figure supplement 1*.

number of Pax7$^+$ MuSCs and the percentage of single MuSCs, and derived clusters and pairs, within each myofibre (*Figure 3E*). Together, these findings clearly candidate the Lnc-Rewind transcript as a regulator of *Wnt7b* repression in muscle stem cells and prompted us towards the study of the underlying mechanism by which this regulation occurs.

## Lnc-Rewind directly interacts with the methyltransferase G9a and mediates specific in cis repression of *Wnt7b* in MuSCs

The above results suggest that Lnc-Rewind exerts a repressive role on *Wnt7b* gene expression. Several works accumulated so far indicate that most of the *cis*-acting chromatin-associated lncRNA function occurs by recruiting and guiding chromatin modifiers to target genes (*Batista and Chang, 2013*; *Guttman and Rinn, 2012*; *Khalil et al., 2009*; *Rinn and Chang, 2012*). In light of our data showing a negative correlation of *Wnt7b* expression by Lnc-Rewind, we focused our attention on the two most known repressive lysine methyltransferases, EZH2 and G9a, which catalyse the deposition of H3K27me3 and H3K9me1/2 on target genes, respectively (*Mozzetta et al., 2015*). Both EZH2 (*Caretti et al., 2004*) and G9a (*Ling et al., 2012*) are mostly expressed in proliferating MuSCs and become downregulated during muscle differentiation (*Figure 4—figure supplement 1A*), similarly to Lnc-Rewind (*Figure 1C*). Moreover, both EZH2 (*Rinn et al., 2007*; *Zhao et al., 2008*) and G9a (*Nagano et al., 2008*; *Pandey et al., 2008*) have been previously reported to be recruited to specific genomic loci through the interaction with different lncRNAs. Thus, we hypothesized that in proliferating myoblasts Lnc-Rewind might interact with EZH2 and/or G9a repressive complexes to tether them on *Wnt7b* genomic locus. Therefore, we performed RNA immunoprecipitation (RIP) analysis in proliferating $C_2C_{12}$ cells using antibodies against G9a and EZH2. We observed that, although both G9a and EZH2 were successfully immunoprecipitated (*Figure 4—figure supplement 1B*, upper panel), Lnc-Rewind transcript was efficiently retrieved only in the G9a native RIP, while it was almost undetectable in EZH2 IP (*Figure 4—figure supplement 1B*, lower panel). To note, the levels of RNA obtained in the specific IP fraction were similar to the ones of the *Kcnq1ot1* lncRNA, which was previously demonstrated to be physically associated with both G9a and Ezh2 proteins (*Pandey et al., 2008*). The specificity of Lnc-Rewind and G9a interaction was strengthened by the use of two reciprocal strategies (*Figure 4*). On the one hand, the G9a-crosslinked immunoprecipitation (CLIP) assay (*Figure 4A*) revealed the presence of Lnc-Rewind in the G9a immunoprecipitated samples. As for RIP, also in this case, Lnc-Rewind RNA-specific enrichment was comparable to the *Kcnq1ot1*-positive control. As a further validation, endogenous Lnc-Rewind RNA pulldown (*Figure 4B*) revealed the presence of G9a in the lncRNA-precipitated fraction. This evidence confirmed the direct interaction between these two partners and led us to hypothesize that Lnc-Rewind might exert its repressive function on the *Wnt7b* locus through G9a. In support of this, depletion of G9a by siRNA-mediated knockdown in MuSC-derived myoblasts induced *Wnt7b* upregulation (*Figure 4C*). To test whether Lnc-Rewind and G9a binds the *Wnt7b* locus in vivo, we applied RNA-DNA-FISH (*Figure 4D*) and a chromatin immunoprecipitation (ChIP) assay (*Figure 4—figure*

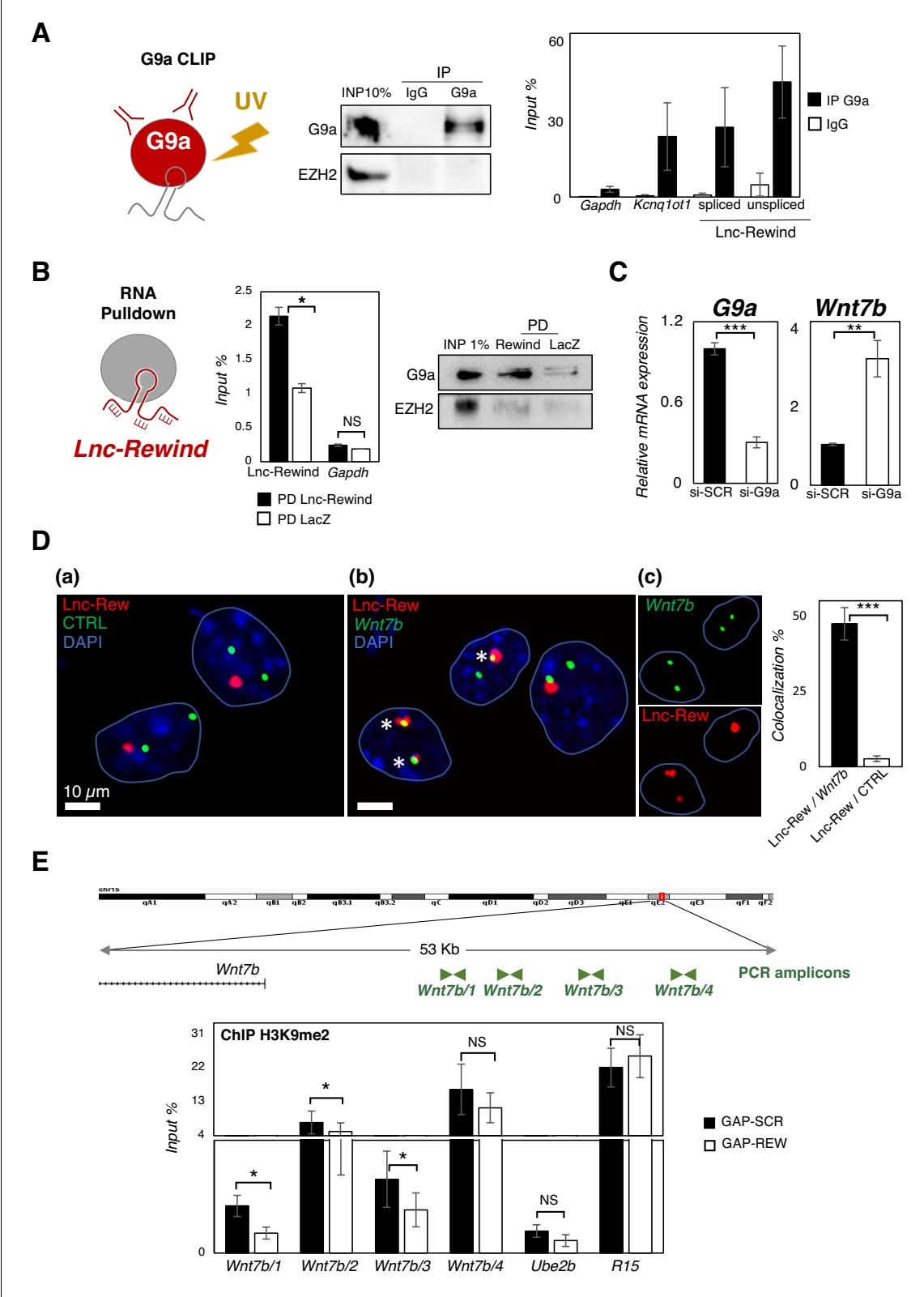

**Figure 4.** Lnc-Rewind directly interacts with the methyltransferase G9a and mediates specific in cis repression of *Wnt7b*. (**A**) G9a crosslinked RNA immunoprecipitation assay (CLIP) performed on nuclear extracts from $C_2C_{12}$ proliferating myoblasts. Western blot analysis of G9a (left panel) and qRT-PCR quantification of Lnc-Rewind recovery (right panel) expressed as input percentage (Input %) are shown. EZH2 protein and *Gapdh* RNA serve as negative controls; *Kcnq1ot* RNA is used as positive control. Data represent mean ± SEM of three biological replicates. (**B**) Lnc-Rewind RNA pulldown

*Figure 4 continued*

assay performed on nuclear extracts from $C_2C_{12}$ proliferating myoblasts. qRT-PCR quantification of Lnc-Rewind recovery (left panel) and Western blot analysis of G9a (right panel) are shown. *Gapdh* RNA and EZH2 protein serve as negative controls. Data represent mean ± SEM of three biological replicates. (C) qRT-PCR quantification of *G9a* and *Wnt7b* in MuSC-derived myoblasts treated with si-SCR or si-G9a. Data were normalized to *Gapdh* mRNA and represent the mean ± SEM of three biological replicates. (D) RNA/DNA-FISH experiments performed in proliferating MuSC-derived myoblasts showing the Lnc-Rewind RNA (red) and a DNA control (CTRL) region (green) (a) or the *Wnt7b* locus (green) (b). Inserts in (c) show single fluorescent channels for Lnc-Rewind and *Wnt7b* signals labelled by asterisks. Blue lines indicate the edges of the nuclei. DAPI, 4′,6-diamidino-2-phenylindole (blue). Histogram reports the mean percentage ± SD of Lnc-Rewind signals colocalizing with *Wnt7b* locus or with the control region from three biological replicates. The extended list of the DNA and RNA probes used is listed in Key resources table. (E) Zoom-in into the genomic region upstream the *Wnt7b* TSS (upper panel). ChIP amplicons used to test the H3K9me2 enrichments are shown in green. Histogram shows the levels of H3K9me2 as analysed by ChIP experiments performed in MuSC-derived myoblasts upon GAP-SCR or GAP-REW transfection (lower panel). The H3K9me2 enrichment is represented as input percentage (Input %). *Ube2b* and *R15* genomic regions were used as negative and positive controls, respectively. The graph shows the mean ± SEM of six independent experiments. Data information: (B) and (E): *p<0.05, paired Student's t-test. (C) and (D): **p<0.01, ***p<0.001, unpaired Student's t-test.

The online version of this article includes the following source data and figure supplement(s) for figure 4:

**Source data 1.** Source data for *Figure 4*.
**Figure supplement 1.** Lnc-Rewind directly interacts with the methyltransferase G9a and mediates specific in cis repression of *Wnt7b*.
**Figure supplement 1—source data 1.** Source data for *Figure 4—figure supplement 1*.

*supplement 1C*), respectively. In agreement with our hypothesis, we found an overlap between Lnc-Rewind (RNA) and *Wnt7b* (DNA) localization (*Figure 4D*) and an intriguing enrichment of the methyltransferase in the *Wnt7b* upstream regions (*Figure 4—figure supplement 1C*). Consistent with a role of Lnc-Rewind in the G9a-mediated epigenetic repression of *Wnt7b*, quantification of the H3K9me2 mark on the G9a-occupied genomic regions (*Figure 4E*, upper panel) showed that H3K9me2 deposition decreased upon Lnc-Rewind knockdown, compared to the GAP-SCR control (*Figure 4E*, lower panel). Moreover, a combined Lnc-Rewind (RNA)/*Wnt7b* (DNA) FISH/G9a (PROTEIN) immunostaining approach showed the colocalization of both G9a and Lnc-Rewind on *Wnt7b* genomic locus (*Figure 5A*). Taken together, these results support a mechanism of action through which in MuSC Lnc-Rewind represses *Wnt7b* gene transcription by mediating the specific G9a-dependent H3K9me2 deposition on its locus (*Figure 5B*).

## Discussion

Wnt signalling represents one of the pathways that has a major role in myogenesis as it is essential for proper MuSC self-renewal and differentiation (*von Maltzahn et al., 2012*). A correct timing of Wnt signalling activation is crucial to obtain proper tissue repair, and its aberrant activity causes a wide range of pathologies (*Nusse and Clevers, 2017*). Thus, it is not surprising that Wnt pathway is under a strict positive and negative multi-layered regulation. Recent studies show that lncRNAs can modulate Wnt pathway by affecting gene expression through different mechanisms, from transcriptional to post-transcriptional level (*Ong et al., 2017*; *Shen et al., 2017*; *Zarkou et al., 2018*) For example, lncRNAs were found interacting with transcription factors (*Desideri et al., 2020*) and chromatin modifiers (*Hu et al., 2015*) altering Wnt signalling pathway in different tissues and in cancer. In this study, we discovered that lncRNAs-mediated regulation plays a role in modulating Wnt pathway in muscle stem cells. Taking advantage of a newly discovered atlas of not annotated muscle-specific lncRNAs (*Ballarino et al., 2015*), we decided to focus our attention on Lnc-Rewind for the following reasons: (1) an orthologue transcript, hs_Lnc-Rewind, with high level of sequence identity is expressed by human myoblasts (*Figure 1B*, *Figure 1—figure supplement 1A,B*); (2) Lnc-Rewind is associated and retained to chromatin (*Figure 1D–F*); (3) it is in genomic proximity to *Wnt7b* locus (*Figure 3A*), and (4) its knockdown induces upregulation of *Wnt7b* in MuSC-derived myoblasts (*Figure 3C*, *Figure 3—figure supplement 1A,B*). Interestingly, the read coverage revealed the existence of multiple isoforms that are not fully spliced. The slowness in the splicing process could contribute to the maintenance of Lnc-Rewind at its own site of transcription improving its retention on the chromatin. In line with this, recent evidence indicates that the presence of introns, and their distinct processing, may represent a way to regulate the nuclear localization and function of many lncRNAs (*Guo et al., 2020*; *Zuckerman and Ulitsky, 2019*). These observations led us hypothesize a

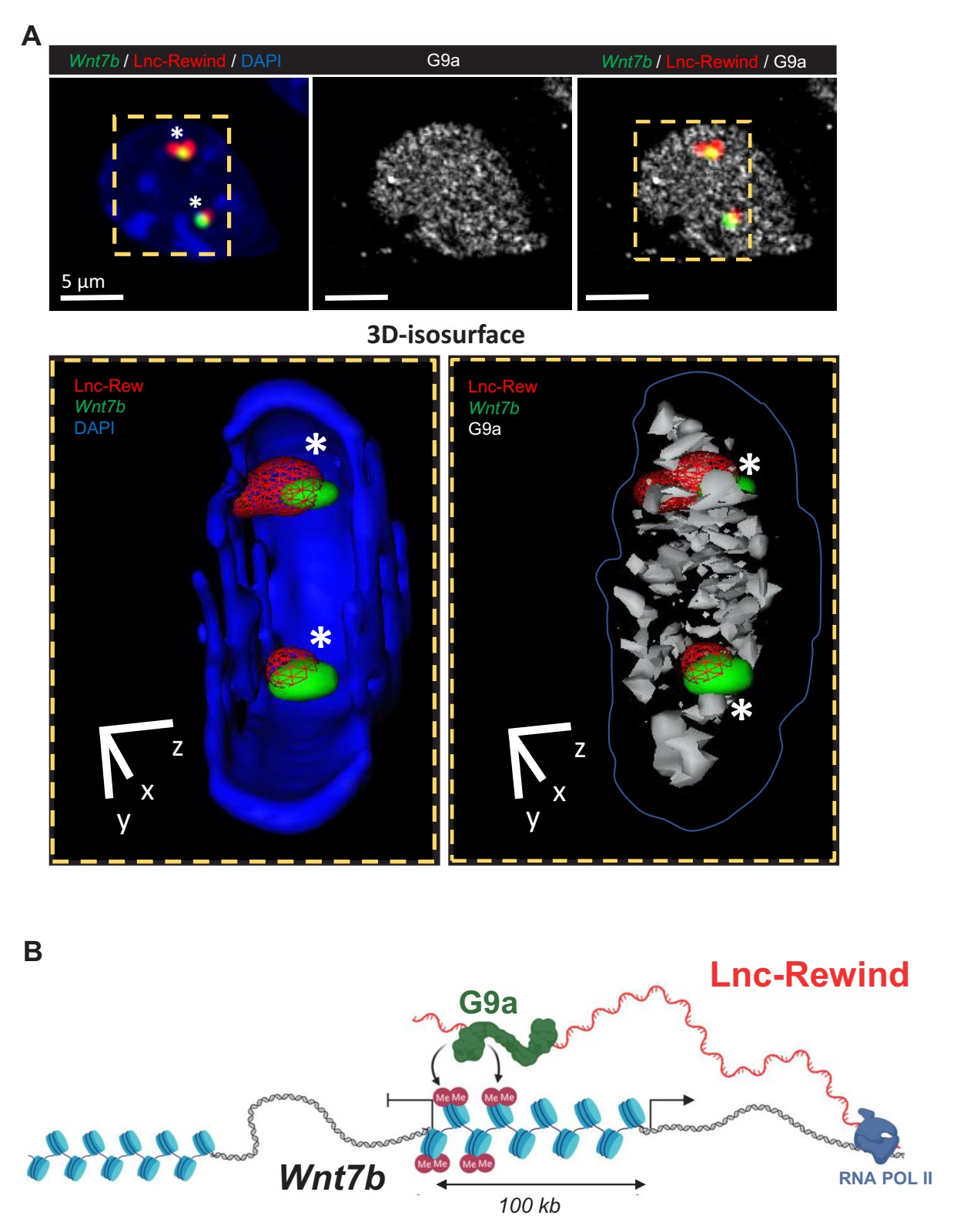

**Figure 5.** Lnc-Rewind and G9a colocalize on *Wnt7b* gene locus. (**A**) Top panel: RNA/DNA-FISH combined with immunofluorescence performed on MuSC-derived myoblasts; Lnc-Rewind RNA (red), *Wnt7b* DNA locus (green), and G9a protein (grey) are shown. Bottom panel: Isosurface rendering of nuclear sections is indicated by the yellow dashed lines. The overlapping regions between Lnc-Rewind (RNA) and *Wnt7b* (DNA) locus (left) and among Lnc-Rewind, *Wnt7b* locus, and G9a (right) are indicated by the white asterisks inside the nuclear volume. (**B**) Proposed model for the functional interplay

*Figure 5 continued on next page*

*Figure 5 continued*

between Lnc-Rewind and G9a on *Wnt7b* gene locus. Representation of mmu_Lnc-Rewind mode of action in muscle cells. In proliferating myoblasts, the Lnc-Rewind transcript is expressed, recruits the silencing methyltransferase G9a on the locus that represses *Wnt7b* transcription.

Lnc-Rewind-mediated in cis repression for *Wnt7b* gene. Of note, by analyzing the expression of surrounding genes within Lnc-Rewind genomic region, *Wnt7b* is the only gene to be significantly upregulated by Lnc-Rewind depletion (*Figure 3C*, *Figure 3—figure supplement 1B*). Moreover, we show that Lnc-Rewind and *Wnt7b* expressions are anti-correlated in different cell types (*Figure 3B*), suggesting that the repressive action of Lnc-Rewind on *Wnt7b* locus could have a more general role, not only in satellite cells.

It is well-accepted that the majority of the *cis*-acting chromatin-associated lncRNAs function by recruiting and targeting chromatin modifiers to specific genes (*Batista and Chang, 2013*; *Guttman and Rinn, 2012*; *Khalil et al., 2009*; *Rinn and Chang, 2012*). In light of the evidence that Lnc-Rewind promotes *Wnt7b* repression, we decided to investigate the involvement of the two major repressive histone modifiers, EZH2 and G9a, in mediating such Lnc-Rewind-dependent silencing. Notably, by performing both protein- and RNA-centric biochemical approaches (RIP/CLIP and RNA pulldown), we found that only G9a specifically interacts with Lnc-Rewind (*Figure 4A, B*, *Figure 4—figure supplement 1B*). This strongly suggests that the histone H3K9 lysine methyltransferase (KMT) might be specifically tethered on *Wnt7b* genomic locus by Lnc-Rewind to mediate its transcriptional repression. In support of this idea, through H3K9me2 ChIP experiments here we provide evidence that different regions upstream *Wnt7b* locus are enriched in the G9a-deposited histone mark in MuSC-derived myoblasts and that H3K9me2 levels on these regions significantly decrease upon Lnc-Rewind knockdown (*Figure 4E*). Moreover, both ChIP (*Figure 4—figure supplement 1C*) and imaging (*Figure 4D*) studies confirmed the presence of G9a in close proximity to *Wnt7b* locus. These data, together with the evidence that G9a downregulation is sufficient to induce *Wnt7b* de-repression (*Figure 4C*), confirm a direct role for G9a-mediated H3K9 methylation in maintaining the repression of *Wnt7b* in MuSCs.

A correct timing and magnitude of Wnt signalling activation is essential to maintain functional MuSCs. For instance, it has been shown that low β-catenin activity is fundamental during early phases of muscle regeneration to allow MuSC activation and subsequent differentiation (*Figeac and Zammit, 2015*; *Parisi et al., 2015*). Accordingly, satellite cells isolated from mice with constitutive active Wnt/β-catenin signalling display an early growth arrest and premature differentiation (*Rudolf et al., 2016*). All these data point out that the cell environment, together with the timing and the proper activation of Wnt signalling pathway, is fundamental for muscle homeostasis.

Although the function of WNT7b has been studied in different cell types and developmental processes (*Chen et al., 2014*; *Wang et al., 2005*), the involvement of this ligand in muscle biology is still rather unexplored. The fact that in MuSCs, *Wnt7b* gene is maintained at very low levels and that it is the latest Wnt ligand induced after muscle injury (*Polesskaya et al., 2003*) suggests the necessity to keep it repressed to allow MuSC expansion. In line with this, here we show that aberrant, cell-autonomous activation of *Wnt7b* expression upon Lnc-Rewind depletion causes defects in MuSC expansion and activation (*Figure 2E–G*) that are abolished if the expression of *Wnt7b* is rescued to physiological levels (*Figure 3D,E*), emphasizing that Lnc-Rewind-mediated repression of *Wnt7b* is key in ensuring MuSC activity. This is nicely supported by a previous study demonstrating that, in the cytoplasm, a *Wnt7b*/lncRNA circuitry controls the proliferation of $C_2C_{12}$ myoblasts. Specifically, Lu and colleagues demonstrated that the YY1-associated muscle lincRNA (*Yam-1*) inhibits skeletal myogenesis through modulation of *Mir715* expression, which in turn targets *Wnt7b* mRNA (*Lu et al., 2013*). In sum, our results support a mechanism of action through which Lnc-Rewind mediates G9a recruitment on *Wnt7b* locus to silence it through the deposition of the repressive mark H3K9me2 during the proliferation phase of satellite cells (*Figure 5B*), thus providing unique insights on the contribution of nuclear-enriched lncRNAs in myogenesis.

# Materials and methods

## Key resources table

| Reagent type (species) or resource | Designation | Source or reference | Identifiers | Additional information |
|---|---|---|---|---|
| Strain, strain background (mouse) | C57Bl/10 (WT) | JAX | Stock# 000665 RRID:MGI:5656893 | |
| Cell line (mouse) | $C_2C_{12}$ | ATCC | C3H RRID:CVCL_UR38 | |
| Cell line (human) | Male myoblast (WT) | Telethon Biobank | N/A | |
| Antibody | Anti-CD31-PE (cell line monoclonal) | MiltenyBiotec | Cat# 130111540 RRID:AB_2657296 | (1:25) |
| Antibody | Anti-CD45-PE (cell line monoclonal) | MiltenyBiotec | Cat# 130110797 RRID:AB_2658218 | (1:25) |
| Antibody | Anti-Ter119-PE (cell line monoclonal) | MiltenyBiotec | Cat# 130112909 RRID:AB_2654115 | (1:25) |
| Antibody | Anti-Sca1-FITC (cell line monoclonal) | MiltenyBiotec | Cat# 130116490 RRID:AB_2751322 | (1:50) |
| Antibody | Anti-$\alpha$7Integrin-APCVio770 (cell line monoclonal) | MiltenyBiotec | Cat# 130095212 Custom | (1:20) |
| Antibody | Anti-G9a (rabbit monoclonal) | Abcam | Cat# ab185050 RRID:AB_2792982 | (CLIP, RIP: 10 ug; ChIP: 5 ug; WB: 1:1000; IF: 1:100) |
| Antibody | Anti-EZH2 (mouse monoclonal) | Cell Signaling | Cat# 3147 RRID:AB_10694383 | (RIP: 5 ug; WB: 1:1000) |
| Antibody | Anti-GAPDH (rabbit polyclonal) | Sigma–Aldrich | Cat# G9545 RRID:AB_796208 | (1:10000) |
| Antibody | Anti-MyHC (mouse monoclonal) | DSHB | Cat# Mf20-s RRID:AB_2147781 | (1:20) |
| Antibody | Anti-CyclinD3 (rabbit polyclonal) | Santa Cruz Biotechnology | Cat# sc-182 RRID:AB_2259653 | (1:200) |
| Antibody | Anti-Wnt7b (rabbit polyclonal) | Abcam | Cat#ab94915 RRID:AB_10675749 | (1:200) |
| Antibody | Anti-$H_3K_9me_2$ (mouse monoclonal) | Abcam | Cat# ab1220 RRID:AB_449854 | (ChIP: 5 ug) |
| Antibody | Anti-Pax7 (mouse monoclonal) | DSHB | Cat# Pax7-s RRID:AB_2299243 | (1:10) |
| Antibody | IgG Anti-Rabbit | Invitrogen | Cat# 14-4616-82 RRID:AB_2865072 | (CLIP, RIP:10 ug; ChIP: 5 ug) |
| Antibody | Donkey anti-rabbit IgG Alexa Fluor Plus 647 | Invitrogen | Cat# A32795 RRID:AB_2762835 | (1:300) |
| Antibody | Anti-Ki67 (rabbit polyclonal) | Abcam | Cat# ab15580 RRID:AB_443209 | (1:100) |
| Sequence-based reagent | siRNA: nontargeting control | Sigma–Aldrich | Cat# SIC007 | |

*Continued on next page*

Continued

| Reagent type (species) or resource | Designation | Source or reference | Identifiers | Additional information |
|---|---|---|---|---|
| Sequence-based reagent | LNA Gapmer: nontargeting control | Qiagen | Cat# 300610 | |
| Sequence-based reagent | Mir-Let7-b | Qiagen | Cat# MS00003122 | |
| Sequence-based reagent | Mir-Let7-c | Qiagen | Cat# 18300 | |
| Sequence-based reagent | siG9a | Sigma–Aldrich | SASI_Mm01_00136174 SASI_Mm_00136174_AS SASI_Mm01_00136175 SASI_Mm01_00136175_AS | |
| Sequence-based reagent | siWnt7b | Sigma–Aldrich | SASI_MM02_00316332 SASI_MM02_00316332_AS SASI_MM01_00033690 SASI_MM01_00033690_AS | |
| Sequence-based reagent | HS-DLC1 mRNA probe set | Advanced Cell Diagnostics, Inc | Ref. 716041 | |
| Sequence-based reagent | Wnt7b BAC probe | Invitrogen clones | RP23-272K17 | |
| Sequence-based reagent | NCTC BAC probe | Invitrogen clones | RP23-352B6 | |
| Sequence-based reagent | Mm-Lnc-Rewind probe set | Advanced Cell Diagnostics, Inc | Ref. 722581 | |
| Chemical compound, drug | TriReagent | Sigma | Cat#T9424 | |
| Chemical compound, drug | DAPI | Sigma | Cat# 28718-90-3 | (1:10000) |
| Chemical compound, drug | DSG (di-succinimidyl glutarate) | Santa Cruz Biotechnology | CAS 79642-50-5 | |
| Commercial assay or kit | SYBR Green Master mix | ThermoFisher Scientific | Cat# A25742 | |
| Commercial assay or kit | Paris Kit | Thermo fisher scientific | Cat# AM1921 | |
| Commercial assay or kit | Direct-Zol RNA MiniPrep Kit | Zymo Research | Cat# R2050 | |
| Commercial assay or kit | MAGnify ChIP | ThermoFisher Scientific | Cat# 492024 | |
| Commercial assay or kit | MyTaq DNA polymerase | Bioline | Cat# bio-21105 | |
| Commercial assay or kit | SuperScript RT Vilo Master Mix | ThermoFisher Scientific | Cat# 11754050 | |
| Commercial assay or kit | BaseScope Reagent Kit v2-RED | Advanced Cell Diagnostics, Inc | Ref. 323900 | |

*Continued*

| Reagent type (species) or resource | Designation | Source or reference | Identifiers | Additional information |
|---|---|---|---|---|
| Commercial assay or kit | MiScript II RT kit | Qiagen | Cat# 218160 | |
| Commercial assay or kit | Lipofectamine 2000 | Invitrogen | Cat# 11668019 | |
| Commercial assay or kit | Click-iT EdU Alexa Flour 594 HCS Assay | Invitrogen | Cat# C10354 | |
| Commercial assay or kit | Pierce ChIP-grade Protein A/G Magnetic Beads | ThermoFisher scientific | Cat# 26162 | |
| Peptide, recombinant protein | Dispase II | Roche | 4942078001 | |
| Peptide, recombinant protein | Proteinase K | Roche | EO0491 | |
| Peptide, recombinant protein | Collagenase A | Roche | 10103578001 | |
| Peptide, recombinant protein | DNase I | Roche | 10104159001 | |
| Peptide, recombinant protein | DNase I | ThermoFisher scientific | #EN0521 | |
| Peptide, recombinant protein | Collagenase I | Sigma | C0130 | |
| Software, algorithm | FACSDiva | BD Biosciences | Version 6.1.3. | |
| Software, algorithm | FlowJo | Tree Star | Version 9.3.2 RRID:SCR_008520 | |
| Software, algorithm | Fiji image processing package | Open-source software (OSS) projects | https://imagej. net/Fiji RRID:SCR_002285 | |
| Software, algorithm | MetaMorph | Molecular Devices | https://www. moleculardevices. com/ RRID:SCR_002368 | |
| Software, algorithm | FV10-ASW Viewer software | Olympus | https://www. olympus-lifescience.com/ RRID:SCR_014215 | |
| Software, algorithm | ZEN 3.0 Blue edition | ZEISS | | |

## Cell preparation and FACS sorting

Cell isolation and labelling were essentially performed as described in *Mozzetta, 2016*. Isolation of cells from 2 months old C57/Bl10 (RRID:MGI:5656893) WT mice was performed as follows: briefly, whole lower hindlimb muscles were carefully isolated, minced, and digested in phosphate-buffered saline (PBS) (Sigma) supplemented with 2.4 U/ml Dispase II (Roche), 100 µg/ml Collagenase A (Roche), 50 mM $CaCl_2$, 1 M $MgCl_2$, 10 mg/ml DNase I (Roche) for 1 hr at 37°C under agitation. Muscle slurries were passed 10 times through a 20G syringe (BD Bioscience). Cell suspension was obtained after three successive cycles of straining and washing in Washing Buffer consisting of HBSS containing 0.2% bovine serum albumin (BSA) (Sigma–Aldrich), 1% penicillin-streptomycin. Cells were

incubated with primary antibodies CD31-PE (MiltenyBiotec, 130111540; RRID:AB_2657296), CD45-PE (MiltenyBiotec, 139110797; RRID:AB_2658218), Ter119-PE (MiltenyBiotec, 130112909; RRID:AB_2654115) 1:25; Sca1-FITC (MiltenyBiotec, 130116490; RRID:AB_2751322) 1:50; α7Integrin-APC-Vio770 (MiltenyBiotec, 130095212; Custom) 1:20 for 45 min on ice diluted in HBSS containing 0.2% BSA, 1% penicillin-streptomycin, and 1% DNAse I. The suspension was finally washed and resuspended in PBS containing 2% fetal bovine serum (FBS) and 0.5 μM ethylenediaminetetraacetic acid (EDTA). Cells were sorted using a FACSAriaIII (Becton Dickinson, BD Biosciences) equipped with 488 nm, 561 nm, and 633 nm laser and FACSDiva software (BD Biosciences, version 6.1.3). Data were analyzed using a FlowJo software (Tree Star, version 9.3.2). Briefly, cells were first gated based on morphology using forward versus side scatter area parameter (FSC-A versus SSC-A) strategy followed by doublets exclusion with morphology parameter area versus width (A versus W). Muscle satellite cells (MuSCs) were isolated as $Ter119_{neg}/CD45_{neg}/CD31_{neg}/a7\text{-}integrin_{pos}/Sca1_{neg}$ cells. To reduce stress, cells were isolated in gentle conditions using a ceramic nozzle of size 100 μm, a low sheath pressure of 19.84 pound-force per square inch (psi) that maintain the sample pressure at 18.96 psi and a maximum acquisition rate of 3000 events/s. Cells were collected in five polypropylene tubes. Following isolation, an aliquot of the sorted cells was evaluated for purity at the same instrument resulting in an enrichment >98–99% (see *Figure 1—figure supplement 1C*, right panel).

## Cell culture conditions and transfection

Freshly sorted cells were plated on ECM Gel (Sigma)-coated dishes in Cyto-grow (Resnova) complete medium as a growth medium (GM) and cultured at 37°C and 5% $CO_2$. After 5 days in GM, MuSC-derived myoblasts were exposed, generally for 2 days, to differentiation medium consisting of Dulbecco's modified Eagle's medium (DMEM) with 5% horse serum (HS). Cells were counted on DAPI-stained images using the Fiji (RRID:SCR_002285) tool 'Multi point'. Downregulation of RNA expression was performed at ~50% confluence by transfection in Lipofectamine 2000 (Invitrogen) according to manufacturer's instructions, with 75 nM of LNA GapmeRs (Euroclone) for the downregulation of Lnc-Rewind and with siRNA (Sigma) for *G9a* and *Wnt7b*, according to manufacturer's instructions. Negative Control A (Euroclone) (GAP-SCR) and Mission siRNA Universal Negative Control (Sigma) (si-SCR) were used as controls. See *Supplementary file 1* and Key resources table for details. MuSC-derived myoblasts were subjected to two consecutive (overnight) rounds of transfection in GM and harvested 24 hr after second transfection.

GapmeRs were designed against Lnc-Rewind sequence using the Exiqon web tool (http://www.exiqon.com/ls/Pages). Negative Control A (Euroclone) was used as negative (Scramble) control. Sequences are reported in *Supplementary file 1* and Key resources table.

For EdU (Invitrogen) detection cells were incubated for 6 hr and stained using Click-iT EdU Alexa Flour 594 HCS Assay (Invitrogen) according to the manufacturer's instructions.

## Single myofibre isolation and immunofluorescence

Single myofibres were isolated from Extensor digitorum longus (EDL) muscles of C57/Bl10 mice and digested in 2 mg/ml collagenase I (Sigma) for 1 hr at 37°C, gently shacked every 10 min. Single fibres were obtained gently triturating the digested EDL muscles using a glass pipette in DMEM supplemented with 10% HS. The myofibres were manually collected under a dissecting microscope and cultured in DMEM + Pyr with 20% FBS, 2.5 ng/ml FGF (Gibco), and 1% chick embryo extract (CEE) (Life Science Production). RNAi experiments were performed 4 hr after plating with one round of transfection, as described above. The fibres were incubated for 24 hr with EdU (Invitrogen) and were analyzed 96 hr after plating.

For EdU detection cells were stained using Click-iT EdU Alexa Flour 594 HCS Assay (Invitrogen) according to the manufacturer's instructions.

For the immunofluorescence, the single myofibres were fixed with 4% paraformaldehyde in PBS for 20 min at room temperature (RT), permeabilized with 0.5% Triton X-100 in PBS, and blocked with 10% FBS in PBS for 1 hr at RT. Primary antibodies (Pax7 (DSHB, Pax7-s; RRID:AB_2299243) 1:10 and Ki67 (Abcam, ab15580; RRID:AB_443209) 1:100) were diluted in 10% FBS–PBS and incubated overnight at 4°C. After incubation for 1 hr with the appropriate secondary antibodies (Alexa Fluor 488 or 594, Thermo Fisher), nuclei were counterstained with DAPI (Sigma), and fibres were mounted

on cover-glasses. Images were taken with Axio Observer microscope (ZEISS) and processed with ZEN 3.0 (Blue edition) software.

## RNA/DNA-FISH

Lnc-Rewind in situ hybridization analyses were performed as previously described (*Rossi et al., 2019*). Briefly, proliferating MuSC-derived myoblasts and $C_2C_{12}$ cells were cultured on pre-coated glass coverslips and then fixed in 4% paraformaldehyde/PBS (Electron Microscopy Sciences, Hatfield, PA). RNA hybridization and signal development were carried out using Basescope assay (Advanced Cell Diagnostics, Bio-Techne) and BA-Mm-Lnc-Rewind probe-set (ref. 722581) designed to detect three junction regions of Lnc-Rewind (exon/intron1, intron1/exon2, exon/intron2). A probe specific for exon junction exon1/exon2 of human Dlc1 mRNA (BA-HS-DLC1, ref. 716041) was used as negative control. Confocal images were acquired at Olympus IX73 spinning disk confocal microscope equipped with a Confocal Imager (CREST X-LIGHT) plus CoolSNAP Myo CCD camera (Photometrics) and at Olympus iX83 FluoView1200 laser scanning confocal microscope. Stacks of images were taken automatically with 0.2 or 0.3 microns between the Z-slices using a 60x NA1.35 oil objective and 405/473/559/635 nm or 405/470/555/640 nm lasers subset, respectively. Filter setting for DAPI, Alexa Fluor 488, Cy3, and Alexa Fluor 594 was used. The images were collected with MetaMorph or FV10-ASW confocal image acquisition software, and post-acquisition processing was performed by FIJI software to the entire image. 3D viewer plugin was used to perform 3D-rendering. DNA-FISH were sequentially carried out after RNA-FISH staining according to *Santini et al., 2021*. In particular, *Wnt7b* and negative control (corresponding to the *nctc* locus, previously used in *Ballarino et al., 2018* and *Desideri et al., 2020*) genomic regions were visualized by nick-translated BAC clones (RP23-272K17 and RP23-352B6, respectively) labelled with Green 496 [5-fluorescein] dUTP (Enzo Life-Sciences). Percentage of colocalized signals in RNA/DNA-FISH experiments were measured as percentage ratio (%) of punctate Lnc-Rewind signals localized on the same focal plane of *Wnt7b* and negative control genomic regions, with respect to the total of Lnc-Rewind signals. In particular, we analysed a total of 135 nuclei for Lnc-Rewind/*Wnt7b* and 126 nuclei for Lnc-Rewind/negative control, from three independent biological replicates. G9a Immunofluorescence was performed sequentially to RNA-FISH staining by incubation in blocking solution (1% goat serum/1% BSA/PBS) for 30 min at RT. G9a primary antibody (Abcam, ab185050) was incubated in 1% donkey serum/PBS overnight at 4°C, while secondary antibody (donkey anti-rabbit Alexa Fluor Plus 647, Invitrogen A32795) were applied in the same incubation buffer for 45 min at RT.

## RNA analyses

Total RNA from myoblasts/myotubes and MuSC-derived myoblasts was extracted with TriReagent (Sigma) using the Direct-Zol RNA MiniPrep Kit (Zymo Research) according to protocol specification. 0.5–1.0 µg of RNA were treated with RNase-free DNase I enzyme (Thermo scientific). Nuclear/cytoplasmic fractionation was performed using the Paris kit (Ambion, AM1921) according to the protocol specifications. RNA was reverse transcribed using the SuperScript VILO Master Mix (Thermo Scientific). Real-time quantitative PCRs were performed by using SYBR Green Master mix (Applied Biosystems), according to the manufacturer's instructions. Relative expression values were normalized to the housekeeping *Gapdh* transcript.

## Western blot

Total proteins were prepared by resuspending cells in RIPA buffer (50 mM Tris–HCl pH 7.4, 150 mM NaCl, 0.1% sodium dodecyl sulphate (SDS), 0.5% sodium deoxycholate, 1% NP-40, 1 mM EDTA, protease and phosphatase inhibitors [Roche]). Protein concentration was determined using a BCA assay (ThermoFisher Scientific). The cell lysate was denatured at 95°C for 5 min. The cell lysates were resolved on 4–15% TGX gradient gels (Bio-Rad Laboratories) and transferred to nitrocellulose membrane (Amersham). Membranes were blocked with 5% non-fat dried milk in TBS with 0.2% Tween for 1 hr at RT and then incubated with primary antibody overnight at 4°C. Primary antibodies used were against MF20 (MyHC) (DSHB, Mf20-s; RRID:AB_2147781), G9a (Abcam, ab185050; RRID:AB_2792982), Ezh2 (Cell Signaling, #3147; RRID:AB_10694383), Wnt7b (Abcam, ab94915; RRID:AB_10675749), Cyclin D3 (Santa Cruz biotechnology, sc-182; RRID:AB_2259653), and GAPDH (Sigma, G9545; RRID:AB_796208). After washing in TBS with 0.2% Tween, membranes were incubated in

HRP-conjugated specific secondary antibody (goat anti-rabbit-anti-mouse [IgG-HRP Santa Cruz Biotechnologies]) for 1 hr at RT. After washing in TBS with 0.2% Tween, blots were developed with Western lightning enhanced chemiluminescence (ThermoFisher Scientific), the signal detection was performed with the use of ChemiDoc (Bio-Rad).

## RNA immunoprecipitation

$C_2C_{12}$ cells were washed with PBS, centrifuged at 2000 rpm for 5 min and resuspended in Buffer A (20 mM Tris–HCl pH 8.0, 10 mM NaCl, 3 mM $MgCl_2$, 0.1% NP40, 10% glycerol, 0.2 mM EDTA, 0.4 mM phenylmethylsulfonyl fluoride (PMSF), 1× Protease Inhibitor cocktail (PIC). After a 15 min incubation on ice, the cell suspension was centrifuged at 2000 rpm for 5 min at 4℃ to pellet the nuclei. The supernatant was collected as cytoplasmic extract. The pellet was re-washed with Buffer A for three times. The pellet was resuspended in NT2/Wash buffer (50 mM Tris pH 7.4, 150 mM NaCl, 1 mM $MgCl_2$, 0.5% NP40, 20 mM EDTA, 1× PIC, 1× phenylmethylsulfonyl fluoride (PMSF), 1 mM dithiothreitol [DTT]), break with 7 ml dounce (tight pestel/B pestel), and centrifuged at 14,000 rpm for 30 min at 4℃. Protein A/G Magnetic beads (Thermo Scientific) were incubate with IgG (Invitrogen, 14-4616-82; RRID:AB_2865072), G9a (Abcam, ab185050; RRID:AB_2792982), and Ezh2 (Cell Signaling, #3147; RRID:AB_10694383) antibody on rotating wheel ON at 4℃, while the nuclear extract (NE) was precleared with the beads. Ten percent of the NE was collected for INP. The NE was divided in each sample with the coated beads (beads + Ab) and incubates on rotating wheel ON at 4℃. The beads were washed with NT2 buffer, and ¼ was collected for protein and ¾ for RNA analyses. RIP qRT-PCR results were represented as percentage of IP/input signal (% input).

## Crosslinking immunoprecipitation

CLIP experiments were performed on NE obtained with some modification of the Rinn et al.'s protocol (*Rinn et al., 2007*). Briefly, $C_2C_{12}$ cells were washed with PBS, crosslinked with UV rays and collected in Buffer A (20 mM Tris–HCl pH 8.0, 10 mM NaCl, 3 mM $MgCl_2$, 0.1% NP40, 10% glycerol, 0.2 mM EDTA, 0.4 mM PMSF, 1× PIC). Cells were centrifuged at 500 x g for 10 min and the supernatant was collected as cytoplasmic extract. The nuclei pellet was re-washed with Buffer A for three times. After the last wash, the nuclei pellet was resuspended in NP40 lysis buffer (50 mM HEPES–KOH, 150 Mm KCl, 2 Mm EDTA, 1 Mm NaF, 0.5% NP40 pH 7.4, 0.5 Mm DTT, 100× PIC), break with 7 ml dounce (tight pestel/B pestel) and centrifuged at maximum speed for 20 min at 4℃. The supernatant was collected and quantified with Bradford assay. Ten percent of the NE was collected for INP. Protein A/G Magnetic beads (Thermo Scientific) were washed with PBS Tween 0.02% (1× PBS, 0.02% Tween-20) and incubated in PBS Tween 0.02% with IgG (Invitrogen, 14-4616-82; RRID:AB_2865072) and G9a (Abcam, ab185050; RRID:AB_2792982) antibody on rotating wheel 1 hr at RT. The NE was divided in each sample with the coated beads (beads + Ab) and incubated on rotating wheel ON at 4℃. The beads were washed three times with the NP40 High Salt Buffer 0.5. (25 mM HEPES–KOH pH7.5, 250 mM KCl, 0.025% NP40) and two times with polynucleotide kinase PNK Buffer (50 mM Tris–HCL, 50 Mm NaCl, 10 mM $MgCl_2$ pH 7.5) and resuspended in 100 μl of NP40 lysis buffer. Seventy-five microlitres were collected for RNA analysis: an equal volume of 2× Proteinase K Buffer (100 mM Tris–HCl, pH 7.5, 150 mM NaCl, 12.5 mM EDTA, 2% [wt/vol] SDS) was added, followed by the addition of Proteinase K (Roche) to a final concentration of 1.2 mg/ml and incubated for 30 min at 55℃. The RNA was recovered and analyzed through qRT-PCR. Twenty-five microlitres were heated at 95℃ for 5 min, and the supernatant collected and resuspended in Protein elution buffer (4× Laemmli sample buffer [Bio-Rad]) with DTT 50 mM and analyzed by Western blot.

## RNA pulldown

RNA pulldown experiments were performed on NE obtained with some modification of the Rinn et al.'s protocol (*Rinn et al., 2007*). $C_2C_{12}$ cells were washed with PBS and harvested in Buffer A (20 mM Tris–HCl pH 8.0, 10 mM NaCl, 3 mM $MgCl_2$, 0.1% NP40, 10% glycerol, 0.2 mM EDTA, 0.4 mM PMSF, 1× PIC). After a 15 min incubation on ice, they were centrifuged at 2000 rpm for 5 min at 4℃ to pellet the nuclei. After three washes performed with Buffer A, the pellet was resuspended in NT2 buffer (50 mM Tris pH7.4, 150 mM NaCl, 1 mM $MgCl_2$, 0.5% NP40, 20 mM EDTA, 1× PIC, 1× PMSF, 1 mM DTT), broken with 1 ml dounce (tight pestel/B pestel) and centrifuged at 14,000 rpm for 30 min at 4℃. The supernatant was quantified with Bradford assay. Streptavidin magnetic beads

(Promega) were incubated with biotinylated primers against Lnc-Rewind and LacZ on a rotating wheel for 30 min at RT, while the NE was precleared with the beads with the same conditions. One percent of the NE was collected for INP. The NE was divided in each sample with the coated beads (beads + primers) and incubated on rotating wheel for 2 hr at RT. The beads were washed with NT2 buffer and 1/5 was collected for the RNA. The remaining 4/5 were eluted in elution buffer (2% SDS, 10% glycerol, 62.5 mM Tris–HCl pH 7.5, 0.5 M DTT, $1\times$ PIC) and incubated 15 min at 70° before incubation of 5 min at 90° in order to collect proteins. Pulldown (PD) qRT-PCR results were represented as percentage of PD/input signal (% input).

## Chromatin immunoprecipitation

ChIP experiments on MuSC-derived myoblasts were performed on chromatin extracts according to the manufacturer's protocol (MAGnify ChIP; Life Technologies) by O.N. incubation with 3 µg of immobilized anti-H3K9me2 (Abcam ab1220; RRID:AB_449854) or rabbit IgG (14-4616-82; RRID:AB_2865072) antibodies. A standard curve was generated for each primer pair testing 5-point dilutions of input sample. Fold enrichment was quantified using qRT-PCR (SYBR Green; Qiagen) and calculated as a percentage of input chromatin (% Inp). Data from GAP-SCR vs GAP-REW conditions represent the mean of six independent experiments ± SEM. For G9a ChIP experiments on proliferating $C_2C_{12}$, crosslinking was performed by adding DSG (di-succinimidyl glutarate; Santa Cruz) at a final concentration of 2 mM for 45 min at RT. Then, formaldehyde (Sigma) was added to culture medium to a final concentration of 1% for 10 min at RT and stopped by glycine to a final concentration of 0.125 M. Chromatin was extracted as described in *Mozzetta et al., 2014* and immunoprecipitated with 5 µg of G9a (Abcam, ab185050; RRID:AB_2792982) or rabbit IgG (14-4616-82; RRID:AB_2865072) antibodies carried out overnight at 4°C. Sequences of the oligonucleotides used for ChIP analyses are reported in *Supplementary file 1*.

## 3′-End mRNA sequencing and bioinformatic analyses

Total RNA was quantified using the Qubit 2.0 Fluorimetric Assay (ThermoFisher Scientific). Libraries were prepared from 100 ng of total RNA using the QuantSeq 3′ mRNA-Seq Library Prep Kit FWD for Illumina (Lexogen GmbH); the quality was assessed by using screen tape high-sensitivity DNA D1000 (Agilent Technologies). Sequencing was performed on a NextSeq 500 using a high-output single-end, 75 cycles, v2 Kit (Illumina Inc). Illumina novaSeq base call (BCL) files were converted in fastq file through bcl2fastq (version v2.20.0.422). Sequence reads were trimmed using trimgalore (v0.4.1) to remove adapter sequences and low-quality end bases (regions with average quality below Phred score 20). Alignment was performed with STAR 2.5.3a (*Dobin et al., 2013*) on mm10 reference. The expression levels of genes were determined with htseq-count 0.9.1 (*Anders et al., 2015*) by using mm10 Ensembl assembly (release 90). Genes having <1 count per million in at least four samples and those with a percentage of multi-mapping reads > 20% were filtered out. Paired t-test was performed to select differentially expressed genes in GAP-REW vs GAP-SCR conditions, setting 0.05 as p-value threshold. Gene Ontology term enrichment analyses were performed using GORILLA (*Eden et al., 2009*) by providing the list of genes expressed in at least one of the two conditions as background.

RNA-Seq reads from WT myoblasts cultured in GM (*Legnini et al., 2017*) were downloaded from GEO (GSE70389), preprocessed using Trimmomatic 0.32 software (*Bolger et al., 2014*), and aligned to human GRCh38 assembly using STAR 2.5.3a. The normalized read coverage tracks (.tdf files) were created and loaded on IGV genome browser (*Robinson et al., 2011*).

## Statistical analyses

The statistical analyses were performed using Microsoft Excel (v16). The use of an unpaired or paired two-tailed Student's t-test is specified in each figure legend. For the data that did not show a normal distribution, the Spearman's rank correlation test was used. When more than two conditions were compared, one-way Anova with Tukey's multiple comparison test was applied using Prism (v9). Statistical significance was set at $p < 0.05$.

## Acknowledgements

The authors acknowledge Marcella Marchioni for technical help. The authors wish to dedicate this manuscript to all researchers, volunteers, and people in healthcare organizations operating around the world to fight coronavirus disease 2019 (COVID-19) infections.

## Additional information

### Funding

| Funder | Grant reference number | Author |
|---|---|---|
| Sapienza Università di Roma | prot. RM11715C7C8176C1 | Monica Ballarino |
| Sapienza Università di Roma | prot. RM11916B7A39DCE5 | Monica Ballarino |
| Ministero dell'Istruzione, dell'Università e della Ricerca | RBSI14QMG0 | Chiara Mozzetta |
| Associazione Italiana per la Ricerca sul Cancro | MyFIRST grant n.18993 | Chiara Mozzetta |
| AFM-Téléthon | #22489 | Chiara Mozzetta |
| Regione Lazio | *POR FESR LAZIO 2014, Project* "RESEARCH" | Chiara Mozzetta |

The funders had no role in study design, data collection and interpretation, or the decision to submit the work for publication.

### Author contributions

Andrea Cipriano, Conceptualization, Data curation, Formal analysis, Supervision, Validation, Investigation, Visualization, Methodology, Writing - review and editing, Performed the cellular and molecular experiments in C2C12 cells, RNA-seq analysis, TSS usage analysis, Gene Ontologies studies, H3K9me2 ChIP experiments and the statistical analyses; Martina Macino, Conceptualization, Data curation, Formal analysis, Validation, Investigation, Visualization, Methodology, Writing - review and editing, performed and analyzed the cellular and molecular experiments carried out on MuSCs and myofibres, RIP, CLIP and G9a ChIP experiments on C2C12, Giulia Buonaiuto, Formal analysis, Validation, Visualization, contributed to the molecular experiments performed in C2C12 and MuSCs cells and H3K9me2 ChIP experiments, performed RNA pull-down; Giulia Buonaiuto, Formal analysis, Validation, Visualization, contributed to the molecular analyses performed in C2C12 and MuSCs cells; Tiziana Santini, Methodology, performed RNA-FISH, RNA/DNA-FISH and RNA/DNA/IF experiments; Beatrice Biferali, Methodology, contributed to the cellular experiments on MuSCs and myofibres; Giovanna Peruzzi, Methodology, perfomed FACS isolation of MuSCs; Alessio Colantoni, Software, performed the bioinformatic analysis of the human myoblast RNA-seq; Chiara Mozzetta, Monica Ballarino, Conceptualization, Resources, Data curation, Supervision, Funding acquisition, Writing - original draft, Writing - review and editing

### Author ORCIDs

Andrea Cipriano https://orcid.org/0000-0003-0775-3198
Martina Macino https://orcid.org/0000-0002-8730-9005
Giulia Buonaiuto https://orcid.org/0000-0002-4099-2433
Beatrice Biferali https://orcid.org/0000-0002-2399-885X
Chiara Mozzetta https://orcid.org/0000-0002-7147-7266
Monica Ballarino https://orcid.org/0000-0002-8595-7105

### Ethics

Animal experimentation: For the experiments described in this study, C57/BL10 wild-type mice were used and differences that were observed in both male and female mice were included in experiments. Animals were treated in respect to housing, nutrition and care according to the guidelines of Good laboratory Practice (GLP). All experimental protocols (Protocol N° 7FF2C.4 -Authorization N°

746/2016-PR) were approved and conformed to the regulatory standards. All animals were kept in a temperature of 22℃ ± 3℃ with a humidity between 50% and 60%, in animal cages with at least five animals.

## Decision letter and Author response
Decision letter https://doi.org/10.7554/eLife.54782.sa1
Author response https://doi.org/10.7554/eLife.54782.sa2

# Additional files

## Supplementary files
• Supplementary file 1. List and sequences of the oligonucleotides, LNA gapmers, and siRNAs used.

• Transparent reporting form

## Data availability
Sequencing data have been deposited in GEO under accession code GSE141396. All data generated or analysed during this study are included in the manuscript and supporting files.

The following dataset was generated:

| Author(s) | Year | Dataset title | Dataset URL | Database and Identifier |
|---|---|---|---|---|
| Cipriano A, Macino M, Buonaiuto G, Santini T, Biferali B, Colantoni A, Peruzzi G, Mozzetta C, Ballarino M | 2021 | 3' mRNA-seq analysis from murine satellite cells upon lnc-Rewind knockdown | https://www.ncbi.nlm.nih.gov/geo/query/acc.cgi?acc=GSE141396 | NCBI Gene Expression Omnibus, GSE141396 |

The following previously published datasets were used:

| Author(s) | Year | Dataset title | Dataset URL | Database and Identifier |
|---|---|---|---|---|
| Ballarino M, Cazzella V, D'Andrea D, Grassi L, Bisceglie L, Cipriano A, Santini T, Pinnarò C, Morlando M, Tramontano A, Bozzoni I | 2014 | Discovery of Novel LncRNA species differentially expressed during murine muscle differentiation | https://www.ebi.ac.uk/ena/browser/view/PRJEB6112 | European Nucleotide Archive (ENA) accession number, PRJEB6112 |
| Legnini I, Briganti F, Sthandier O, Bozzoni I | 2016 | Gene expression profiling of human and murine in vitro muscle differentiation | https://www.ncbi.nlm.nih.gov/geo/query/acc.cgi?acc=GSE70389 | NCBI Gene Expression Omnibus, GSE70389 |
| Arner E, Daub CO, Vitting-Seerup K, Andersson R, Lilje B, Drablos F, Lennartsson A, Rönnerblad M, Hrydziuszko O, Vitezic M, Freeman CT, Alhendi A, Arner P, Axton R, Baillie JK, Beckhouse A, Bodega B, Briggs J, Brombacher F, Davis M, Detmar M, Ehrlund A, Endoh M, Eslami A, Fagiolini M, | 2015 | Transcribed enhancers lead waves of coordinated transcription in transitioning mammalian cells | https://fantom.gsc.riken.jp/5/data/ | FANTOM5 Consortium, fantom |

Fairbairn L, Faulkner GJ, Ferrai C, Fisher ME, Forrester L, Goldowitz D, Guler R, Ha T, Hara M, Herlyn M, Ikawa T, Kai C, Kawamoto H, Khachigian L, Klinken PS, Kojima S, Koseki H, Klein S, Mejhert N, Miyaguchi K, Mizuno Y, Morimoto M, Morris KJ, Mummery C, Nakachi Y, Ogishima S, Okada-Hatakeyama M, Okazaki Y, Orlando V, Ovchinnikov D, Passier R, Patrikakis M, Pombo A, Qin X-Y, Roy S, Sato H, Savvi S, Saxena A, Schwegmann A, Sugiyama D, Swoboda R, Tanaka H, Tomoiu A, Winteringham LN, Wolvetang E, Yanagi-Mizuochi C, Yoneda M, Zabierowski S, Zhang P, Abugessaisa I, Bertin N, Diehl AD, Fukuda S, Furuno M, Harshbarger J, Hasegawa A, Hori F, Ishikawa-Kato S, Ishizu Y, Itoh M, Kawashima T, Kojima M, Kondo N, Lizio M, Meehan TF, Mungall CJ, Murata M, Nishiyori-Sueki H, Sahin S, Sato-Nagao S, Severin J, Kawai J, Kasukawa T, Lassmann T, Suzuki H, Kawaji H, Summers KM, Wells C, FANTOM Consortium, Hume DA, Forrest ARR, Sandelin A, Carninci P, Hayashizaki Y

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
