## [Decision Letter]

Thank you for submitting your article "Epigenetic regulation of Wnt7b expression by the cis-acting long noncoding RNA lnc-Rewind in muscle stem cells" for consideration by *eLife*. Your article has been reviewed by three peer reviewers, and the evaluation has been overseen by a Senior Editor, a Reviewing Editor, and three reviewers. The following individual involved in review of your submission has agreed to reveal their identity: Ahmad S Khalil (Reviewer #2).

The reviewers have discussed the reviews with one another and the Editors. We will not be able to offer publication of the manuscript, at least not in the present form. You'll see that the reviewers have a number of criticisms and some feel that the data are of a preliminary quality that would need to be improved through additional experiments. While we cannot accept the current manuscript, we remain open to receiving a much revised version once you have had a chance to address the substantial concerns.

Summary:

LncRNAs serve as key regulators of gene expression and have recently been demonstrated to be involved in skeletal muscle development and disease. In this manuscript, Cipriano et al., characterize a novel lncRNA, lnc-Rewind, that acts a regulator of muscle stem cell proliferation through a cis-acting chromatin regulation mechanism. Using experiments with mouse cell lines and primary cells, they propose that lnc-Rewind is a conserved chromatin-associated lncRNA, whose expression is associated with muscle stem cell proliferation, that directly interacts with the methyltransferase G9a to mediate repression of Wnt7b.

Essential revisions:

1) Please provide direct evidence that lnc-Rewind interacts with the Wnt7b locus in cis. It would be ideal to perform a CHART/ChIRP/RAP experiment to determine if lnc-Rewind interacts with the Wnt7b locus. An RNA-DNA FISH would also be good to perform to determine the behavior of the RNA relative to the Wnt7b locus. Is the RNA confined to the locus or does it diffuse?

2) It is also not clear that lnc-Rewind directly interacts with G9a. Biochemical data will be required to demonstrate an interaction.

3) If lnc-Rewind upregulates 332 and downregulates 756 via Wnt7b derepression, an essential experiment would be to determine if Wnt7a overexpression in muscle cells results in a similar expression change.

4) Reviewer 3 questions how important Lnc-Rewind is to muscle homeostasis and differentiation; and how important is Wnt7b for the observed phenotype? Additional evidence to support the claims would be essential.

5) The percent of Lnc-Rewind precipitated with antibodies directed to G9a is very low. How much of the known G9a target Kcnq1ot1 precipitated in this assay? Is this background or a specific signal? More controls are required.

6) Furthermore, the H3K9me3 ChIP shows small differences between GAP-SCR (control) and GAP-REW (experiment), and the regions that were analyzed are located about 17-28 kb away from Wnt7b. We agree with Reviewer 3 that a ChIP with anti G9a would have been more informative, and a 4C analysis might show a connection between regions 91-94 and the Wnt7b gene.

---

## [Author Response]

Essential revisions:1) Please provide direct evidence that lnc-Rewind interacts with the Wnt7b locus in cis. It would be ideal to perform a CHART/ChIRP/RAP experiment to determine if lnc-Rewind interacts with the Wnt7b locus. An RNA-DNA FISH would also be good to perform to determine the behavior of the RNA relative to the Wnt7b locus. Is the RNA confined to the locus or does it diffuse?

Our initial RNA-FISH analyses evidenced a pattern of Lnc-Rewind fluorescence which is quite confined to discrete spots rather than being diffused through the nucleoplasm (Figure 1E and Figure 1—figure supplement 1F and 1G). These findings, together with the genomic proximity between *Lnc-Rewind* and *Wnt7b* genomic loci, gave a first piece of advice into a local, *cis*-acting, regulatory mechanism trough which Lnc-Rewind might modulate *Wnt7b* expression. In agreement with the need to provide a more direct evidence on the proposed *in cis* interaction, we have performed a double RNA/DNA FISH approach. Differently from CHART/ChIRP/RAP experiments, which are considered as population-based approaches, this assay allows a quantification of possible RNA/DNA proximities at single cell level. For this, we have labeled Lnc-Rewind RNA (BaseScopeTM technology) and *Wnt7b* locus (BAC clone Catalog#: RP23-272K17) with different fluorophores and used these FISH probes in MuSCs-derived myoblasts. Signal acquisition by confocal microscopy evidenced that ~47% of the Lnc-Rewind nuclear signals colocalize with *Wnt7b* locus (revised Figure 4D(b)). No proximity was instead found when an unrelated region (BAC clone, Catalog#: RP23-352B6 chromosome 7) was used as negative control (revised Figure 4D(a)). Importantly, in the revised version, we have also strengthened the evidence on the concomitant recruitment of Lnc-Rewind and G9a on *Wnt7b*. These data, which required the use of a new imaging set-up able to combine the RNA/DNA-FISH signals with the G9a immunofluorescence, clearly show the close proximity of both Lnc-Rewind and G9a signals with *Wnt7b* gene locus (revised Figure 5A). Overall, we believe that the dissection of the Lnc-Rewind mechanism of action has been considerably improved as a result of these revisions in further supporting its *cis*-acting role into the regulation of Wnt7b expression.

2) It is also not clear that lnc-Rewind directly interacts with G9a. Biochemical data will be required to demonstrate an interaction.

We thank the reviewer for having raised this point on the direct interaction between Lnc-Rewind and G9a whose clarification improved our experimental strategy and the general quality of the manuscript. Several evidences available in literature highlight the high rate of false positives identified by native IP analyses as many artificial interactions between RNA and proteins can occur in vitro after cell lysis (Colantoni et al., 2020; Cipriano et al., 2018; McHugh et al., 2014). Being aware of this limitation, in the previous version of the paper we already combined our native approach (RIP, Figure 4—figure supplement 1B) with another, crosslinked-based, protein-centric strategy (CLIP, revised Figure 4A). Please, kindly note that the CLIP, assay, which confirmed the specificity of Lnc-Rewind interaction with G9a, is recognized as the safest approach to detect *direct* interactions between RNA and proteins as the UV wavelength (254nm) is able to induce the crosslinking only if the RNA and the protein are in close contact (Ramanathan, et al., 2019). Based on the reviewers’ concern on the need of other biochemical data, we further applied an RNA pull-down approach allowing the purification of the *endogenous* Lnc-Rewind RNA and its interacting proteins. This analysis, which is RNA-centric and complementary to our initial RIP and CLIP experiments, strengthen our initial evidence by confirming the presence of G9a in Lnc-Rewind precipitated sample. These new results have been incorporated the revised Figure 4B and discussed in the subsection “Lnc-Rewind directly interacts with the methyltransferase G9a and mediates specific *in cis* repression of *Wnt7b* in MuSCs”. Additional support was provided, as discussed above, by the RNA/DNA-FISH/IF analysis showing a G9a/Lnc-Rewind concomitant recruitment on *Wnt7b* genomic locus (revised Figure 5A).

3) If lnc-Rewind upregulates 332 and downregulates 756 via Wnt7b derepression, an essential experiment would be to determine if Wnt7a overexpression in muscle cells results in a similar expression change.

While it is true that the expression of several genes is affected upon Lnc-Rewind knockdown, our transcriptomic data are not hierarchical and do not assume Wnt7b as the *primum movens* of the regulatory cascade downstream to Lnc-Rewind. Although some overlapping might exist, Wnt7b overexpression in muscle cells is not expected to globally modulate the same subset of Lnc-Rewind target genes. An additional issue is that Wnt7b ectopic expression will not necessarily mimic the overexpression level obtained upon Lnc-Rewind knockdown. It is well known that different levels of Wnt pathway activation give rise to diverse, and sometime opposite, phenotypes (Rudolf et al., 2016Β; Parisi et al., 2015; Figeac and Zammit, 2015). Therefore, unbalanced levels of Wnt7b overexpression might give rise to contradictory results. Nonetheless, to link the phenotype observed by Lnc-Rewind depletion to the induction of Wnt7b expression, we considered more straightforward to perform a rescue experiment. Thus, within a contest where the upregulation of Wnt7b was triggered by Lnc-Rewind depletion, we interfered with Wnt7b expression by siRNA-based knockdown. This experiment allowed us to show that restoration of Wnt7b to physiological levels is able to rescue the proliferation defects observed upon depletion of Lnc-Rewind. These results add an important piece of evidence supporting the role of Lnc-Rewind in sustaining MuSCs proliferation by controlling the expression of Wnt7b. These data are now discussed in subsection “Lnc-Rewind regulates muscle system processes and MuSCs” and included in the revised figures (revised Figure 3D,E and new Figure 3—figure supplement 1C).

4) Reviewer 3 questions how important Lnc-Rewind is to muscle homeostasis and differentiation; and how important is Wnt7b for the observed phenotype? Additional evidence to support the claims would be essential.

We agree with the reviewer on the importance of clarifying the impact of Wnt7b on the observed phenotype and we thank him/her for the input to provide additional evidence to strengthen our conclusion. As also described above, to fulfil reviewer’s request we performed a rescue experiment depleting Wnt7b expression upon Lnc-Rewind KD-induced upregulation (new Figure 3—figure supplement 1C). As shown in Figure 3E, this approach allowed us to uniquely demonstrate that the proliferation/activation defects observed on MuSCs depleted of Lnc-Rewind were rescued by restoring Wnt7b expression to control levels, through si-Wnt7b treatment. Of note, we proved this on both MuSCs-derived myoblasts (revised Figure 3D) and in myofibers-associated MuSCs (revised Figure 3E), consolidating our conclusion on a direct role of Lnc-Rewind-mediated Wnt7b repression in sustaining MuSCs proliferation and activation. These data are now discussed in subsection “Lnc-Rewind regulates muscle system processes and MuSCs”.

5) The percent of Lnc-Rewind precipitated with antibodies directed to G9a is very low. How much of the known G9a target Kcnq1ot1 precipitated in this assay? Is this background or a specific signal? More controls are required.

The low percentage of Lnc-Rewind precipitation might be imputable to a general limitation of native RIP. Please kindly note that, indeed, higher levels of enrichments were obtained by CLIP assay (25% of the input, revised Figure 4A, right panel) when the interaction between Lnc-Rewind and G9a was stabilized by UV-cross-linking. As requested, we have quantified the levels of Kcnq1ot1 precipitation and found similar levels of Lnc-Rewind enrichment (see new Figure 4—figure supplement 1B). Together with the biochemical experiments discussed at point *#*2, the above results unequivocally confirm a direct and specific interaction between Lnc-Rewind and G9a. We thank the reviewer for his/her pertinent comment, which has helped to improve our data and to clarify the details of the proposed model.

6) Furthermore, the H3K9me3 ChIP shows small differences between GAP-SCR (control) and GAP-REW (experiment), and the regions that were analyzed are located about 17-28 kb away from Wnt7b. We agree with Reviewer 3 that a ChIP with anti G9a would have been more informative, and a 4C analysis might show a connection between regions 91-94 and the Wnt7b gene.

The choice for these regions was driven by data already available in Mozzetta’s lab, and derived from G9a ChIP-seq analyses performed in C_2_C_12_ (not shown), which revealed the presence of several G9a enriched peaks upstream *Wnt7b*, in particular on the region that is proximal to TSS. In agreement with the need to better specify the reason why these amplicons were prioritized, we have included the G9a ChIP analyses in the revised version (see new Figure 4—figure supplement 1C ) and showed by qPCR that (i) these regions are significantly enriched in G9a and that (ii) at these sites, the levels of G9a enrichment was even higher than *Myogenin*, known G9a-target (Ling et al., 2012). In further support of a role for G9a in mediating Wnt7b repression, we also provide evidence that G9a knock-down induces upregulation of *Wnt7b* mRNA levels on MuSCs. These new data are shown in the revised Figure 4C and commented in subsection “Lnc-Rewind directly interacts with the methyltransferase G9a and mediates specific *in cis* repression of *Wnt7b* in MuSCs”.